# Barriers to engagement in the care cascade for tuberculosis disease in India: A systematic review of quantitative studies

Tulip A. Jhaveri[1,2], Disha Jhaveri[3,4], Amith Galivanche[3], Maya Lubeck-Schricker[3], Dominic Voehler[3], Mei Chung[3,5], Pruthu Thekkur[6,7], Vineet Chadha[8], Ruvandhi Nathavitharana[9], Ajay M. V. Kumar[6,7,10], Hemant Deepak Shewade[11], Katherine Powers[3], Kenneth H. Mayer[9,12], Jessica E. Haberer[13], Paul Bain[14], Madhukar Pai[15], Srinath Satyanarayana[6,7], Ramnath Subbaraman[2,3]*

1 Division of Infectious Diseases, University of Mississippi Medical Center, Jackson, Mississippi, United States of America, 2 Division of Geographic Medicine and Infectious Diseases, Tufts Medical Center, Boston, Massachusetts, United States of America, 3 Department of Public Health and Community Medicine and Center for Global Public Health, Tufts University School of Medicine, Boston, Massachusetts, United States of America, 4 Harvard T.H. Chan School of Public Health, Boston, Massachusetts, United States of America, 5 Friedman School of Nutrition Science and Policy, Tufts University, Boston, Massachusetts, United States of America, 6 Centre for Operational Research, International Union Against Tuberculosis and Lung Disease (The Union), Paris, France, 7 South-East Asia Office, International Union Against Tuberculosis and Lung Disease (The Union), New Delhi, India, 8 National Tuberculosis Institute, Bengaluru, India, 9 Division of Infectious Diseases, Beth Israel Deaconess Medical Center and Harvard Medical School, Boston, Massachusetts, United States of America, 10 Department of Community Medicine, Yenepoya Medical College, Yenepoya (deemed to be university), Mangalore, India, 11 Division of Health Systems Research, ICMR-National Institute of Epidemiology, Chennai, India, 12 The Fenway Institute, Boston, Massachusetts, United States of America, 13 Center for Global Health, Massachusetts General Hospital and Harvard Medical School, Boston, Massachusetts, United States of America, 14 Countway Library of Medicine, Boston, Massachusetts, United States of America, 15 Department of Global and Public Health and McGill International TB Centre, McGill University, Montreal, Canada

* ramnath.subbaraman@tufts.edu

**Data Availability Statement:** All relevant data are within the manuscript and its Supporting Information files.

## Abstract

### Background

India accounts for about one-quarter of people contracting tuberculosis (TB) disease annually and nearly one-third of TB deaths globally. Many Indians do not navigate all care cascade stages to receive TB treatment and achieve recurrence-free survival. Guided by a population/exposure/comparison/outcomes (PECO) framework, we report findings of a systematic review to identify factors contributing to unfavorable outcomes across each care cascade gap for TB disease in India.

### Methods and findings

We defined care cascade gaps as comprising people with confirmed or presumptive TB who did not: start the TB diagnostic workup (Gap 1), complete the workup (Gap 2), start treatment (Gap 3), achieve treatment success (Gap 4), or achieve TB recurrence-free survival (Gap 5). Three systematic searches of PubMed, Embase, and Web of Science from January 1, 2000 to August 14, 2023 were conducted. We identified articles evaluating

**Funding:** This manuscript was supported by a Doris Duke Clinical Scientist Development Award (grant 2018095) and a Doris Duke Data Sharing Award (grant 2021074), both to RS from the Doris Duke Foundation (https://www.dorisduke.org/). Additional support was provided by a grant from the Bill & Melinda Gates Foundation (grant INV-038215; https://www.gatesfoundation.org). The funders had no role in study design, data collection and analysis, decision to publish, or preparation of the manuscript.

**Competing interests:** MP is an Academic Editor on PLOS Medicine's editorial board, and serves as Editor-in-Chief of PLOS Global Public Health. MP also serves as an advisor to the following non-profit agencies in global health: Bill & Melinda Gates Foundation; Foundation for Innovative New Diagnostics; World Health Organization & the Stop TB Partnership. JEH has been a paid consultant for Merck. JEH owns stock in Natera. All other authors have declared that no competing interests exist.

**Abbreviations:** BMI, body mass index; CI, confidence interval; DOTS, directly observed therapy short course; HIV, human immunodeficiency virus; NTEP, National TB Elimination Programme; MDR, multidrug-resistant; PECO, population/exposure/comparison/outcomes; PTLFU, pretreatment loss to follow-up; RR, rifampin-resistant; TB, tuberculosis.

factors associated with unfavorable outcomes for each gap (reported as adjusted odds, relative risk, or hazard ratios) and, among people experiencing unfavorable outcomes, reasons for these outcomes (reported as proportions), with specific quality or risk of bias criteria for each gap. Findings were organized into person-, family-, and society-, or health system-related factors, using a social-ecological framework. Factors associated with unfavorable outcomes across multiple cascade stages included: male sex, older age, poverty-related factors, lower symptom severity or duration, undernutrition, alcohol use, smoking, and distrust of (or dissatisfaction with) health services. People previously treated for TB were more likely to seek care and engage in the diagnostic workup (Gaps 1 and 2) but more likely to suffer pretreatment loss to follow-up (Gap 3) and unfavorable treatment outcomes (Gap 4), especially those who were lost to follow-up during their prior treatment. For individual care cascade gaps, multiple studies highlighted lack of TB knowledge and structural barriers (e.g., transportation challenges) as contributing to lack of care-seeking for TB symptoms (Gap 1, 14 studies); lack of access to diagnostics (e.g., X-ray), non-identification of eligible people for testing, and failure of providers to communicate concern for TB as contributing to non-completion of the diagnostic workup (Gap 2, 17 studies); stigma, poor recording of patient contact information by providers, and early death from diagnostic delays as contributing to pretreatment loss to follow-up (Gap 3, 15 studies); and lack of TB knowledge, stigma, depression, and medication adverse effects as contributing to unfavorable treatment outcomes (Gap 4, 86 studies). Medication nonadherence contributed to unfavorable treatment outcomes (Gap 4) and TB recurrence (Gap 5, 14 studies). Limitations include lack of meta-analyses due to the heterogeneity of findings and limited generalizability to some Indian regions, given the country's diverse population.

## Conclusions

This systematic review illuminates common patterns of risk that shape outcomes for Indians with TB, while highlighting knowledge gaps—particularly regarding TB care for children or in the private sector—to guide future research. Findings may inform targeting of support services to people with TB who have higher risk of poor outcomes and inform multicomponent interventions to close gaps in the care cascade.

## Author summary

### Why was this study done?

- India has the highest tuberculosis (TB) incidence, accounting for about one-quarter of people with TB disease and nearly one-third of TB deaths globally.

- Many Indians with TB do not traverse all care stages needed to receive treatment and achieve an optimal long-term outcome, with serial losses of people across these stages referred to as the "care cascade."

- Understanding why losses of people with TB disease occur across the care cascade is crucial to inform interventions to prevent unfavorable outcomes.

**What did the researchers do and find?**

- We conducted 3 systematic searches to identify papers published from 2000 to 2023.

- We extracted information from these studies on risk factors for unfavorable outcomes for each care cascade gap, as well as reasons reported by people with TB who experienced unfavorable outcomes and were surveyed by researchers.

- Some factors contributed to losses at multiple care cascade stages, including male sex, older age, poverty-related factors, history of prior TB treatment, lower symptom severity or duration, undernutrition, alcohol use, smoking, and dissatisfaction with health services.

- Other barriers included: lack of TB knowledge and transportation barriers to clinic contributing to lack of care-seeking (Gap 1), poor accessibility of testing and failure to identify people eligible for testing contributing to non-completion of the diagnostic workup (Gap 2), early deaths from diagnostic delays and poor recording of contact information contributing to losses of people before treatment (Gap 3), lack of TB knowledge and depression contributing to unfavorable treatment outcomes (Gap 4), and medication nonadherence contributing to unfavorable treatment outcomes and TB recurrence (Gaps 4 and 5).

**What do these findings mean?**

- Reasons for losses of people with TB disease across the care cascade are complex, vary by care cascade gap, and involve patient- and health system-related barriers.

- India's TB program should target additional services to people with higher risk of poor outcomes and develop multicomponent interventions to address the diverse challenges faced by people with TB.

- Study limitations include lack of meta-analyses (i.e., estimation of the average effect of each risk factor by combining findings across studies), and caution is required when applying findings across India's diverse population.

## Introduction

With an incidence of 2.8 million people with tuberculosis (TB) disease in 2022, India accounts for about one-quarter of people contracting TB and nearly one-third of TB deaths globally [1]. India has also historically had many "missing" people with TB, individuals not reported to the National TB Elimination Programme (NTEP), who may not have received effective care [2].

Losses of people with a disease across sequential care stages needed to achieve a favorable health outcome may be represented using care cascades (or continuums) [3,4]. TB care cascade analyses for India and other countries have provided insights into shortcomings in quality of care [5–9]. For example, although TB programs have historically focused on improving treatment outcomes, in India's NTEP, comparable or greater losses occur during the diagnostic workup, during linkage to treatment, and due to TB recurrence [5]. Based on these insights, India's National Strategic Plan for TB (2017–2025) emphasizes the importance of reducing

care cascade losses to achieve the 2030 World Health Organization (WHO) End TB targets [10,11]. While prior TB care cascade analyses quantified gaps in care, few analyses have mapped findings from studies to understand *who* is lost and *why* people are lost across care cascade stages [12].

In this paper, we report findings of a systematic review of more than 2 decades of quantitative literature on barriers contributing to unfavorable outcomes across India's TB care cascade. While factors vary across India's diverse population, identifying common challenges may guide interventions at the local and national levels, because TB care in India is informed by uniform guidelines (for the public sector [10,13]) and standards (for the private sector [14]). This review aims to inform interventions across care cascade stages to improve the lives of people with TB and accelerate TB elimination in the world's largest epidemic [12].

## Methods

### TB care cascade framework

This review expands upon a prior systematic review, conducted by some of the authors of this paper, estimating India's TB care cascade [5]. While that prior review estimated outcomes in the care cascade, this paper extracts separate findings regarding exposures influencing TB care cascade outcomes. Methods are informed by TB care cascade guidelines [3]. Our review spans 5 care cascade gaps (Table 1), involving 3 search strategies registered in PROSPERO in April 2020. Protocols for each gap are in the S1–S5 Appendices, following the Preferred Reporting Items for Systematic Reviews and Meta-Analyses (PRISMA) guideline (S7 Appendix Checklist).

### PECO framework

Using a population/exposures/comparisons/outcomes (PECO) framework [16], the population comprised people with confirmed or presumptive TB disease in the public or private sector. The directly observed therapy short course (DOTS) strategy expanded starting in 1997 to cover most of India with public services by the early 2000s [17]. Since 2012, the NTEP has mandated private sector TB reporting and provided support through public-private initiatives [18,19].

We extracted data on exposures contributing to poor outcomes for each care cascade gap. Exposures comprised findings from 2 study designs. The first set of findings—referred to as "factors associated with unfavorable outcomes"—derived from cross-sectional, case-control, cohort, and experimental studies comparing people who completed a cascade stage to those who did not. Studies reported findings for exposures as odds, relative risk, or hazard ratios. Comparison groups depended on the exposure—e.g., for male sex, female sex was the reference group.

The second set of findings—referred to as "reasons reported for unfavorable outcomes"— were from studies that surveyed people with unfavorable outcomes. Surveys assessed why people with TB symptoms had not sought care (for Gap 1) or completed the diagnostic workup (Gap 2), or why people diagnosed with TB had not started treatment (for Gap 3) or achieved treatment success (for Gap 4). Studies described the proportion of people reporting specific reasons for unfavorable outcomes, without a reference group.

Outcomes were guided by definitions of each care cascade gap (Table 1 and S1–S5 Appendices). For example, Gap 1 studies evaluated people with TB symptoms in the community who had or had not sought care when the survey was conducted, as a proxy for understanding the behavior of people with undiagnosed TB in the community.

**Table 1. Definitions for TB care cascade gaps, populations, and outcomes examined in this systematic review [3].**

| Care cascade gap | Definition | Populations and outcomes assessed |
|---|---|---|
| Gap 1 | People with symptoms of TB disease in the community who did not reach care and start the diagnostic workup | We included studies of people in the community identified through cross-sectional, population-based surveys, who had TB symptoms. Because these surveys started with symptom screening, studies only included symptomatic individuals. |
| Gap 2 | People with symptoms of TB disease who started but did not complete the diagnostic workup | Given changes in diagnostic algorithms over time, we report findings by non-completion of each diagnostic modality, including non-pursual of TB workup despite referral, non-completion of sputum microscopy, non-completion of chest X-ray, and non-completion of NAAT (e.g., Xpert MTB/RIF, Truenat). |
| Gap 3 | People who were diagnosed with TB disease but did not start or get registered for treatment | We disaggregate findings by people with drug-susceptible TB, people with drug-resistant TB, and children with TB. |
| Gap 4 | People who started treatment but did not achieve treatment success | We disaggregate findings into people with drug-susceptible TB (including those with new TB or a prior TB treatment history), people with drug-resistant TB (including those with isoniazid monoresistance, rifampin resistance, or multidrug resistance), people with HIV being treated for TB, and children with TB. WHO-defined unfavorable outcomes included death, treatment failure, loss to follow-up, or non-evaluation (i.e., not reported, transferred out or treatment regimen modified), alone or in combination. Medication nonadherence was also included as an unfavorable outcome [15]. |
| Gap 5 | People who achieved TB treatment success, but experienced posttreatment disease recurrence or death | We disaggregated studies evaluating TB recurrence versus posttreatment mortality alone. Outcomes could have been reported alone or as part of a composite outcome, along with on-treatment outcomes. |

HIV, human immunodeficiency virus; NAAT, nucleic acid amplification testing; TB, tuberculosis; WHO, World Health Organization.

## Search strategy and study selection

Librarians conducted 3 sets of initial searches (i.e., separate search strategies for Gap 1; Gaps 2, 3, and 4 together; and Gap 5) and 2 refresher searches of PubMed, Embase, and Web of Science using terms in Table A in each of the S1–S5 Appendices. Searches collectively spanned January 1, 2000 (when India's TB program was achieving national coverage [17]) to August 14, 2023. Studies were also identified by reviewing references of included studies and outreach to experts. While searches were not restricted by language, all studies meeting inclusion criteria were published in English.

Using Covidence software (Veritas Health Innovation, Australia), 2 reviewers (from TJ, DJ, AG, DV, or KP) evaluated each article for eligibility at the title and abstract and full-text stages. Disagreements were resolved by a third reviewer (RS). PRISMA flowcharts are in Fig A in each of the S1–S5 Appendices. While inclusion and exclusion criteria varied by gap, in general, included studies compared people who did or did not complete a TB care cascade stage (i.e., factors associated with unfavorable outcomes) or surveyed people who experienced unfavorable outcomes (i.e., reasons reported for unfavorable outcomes). We excluded clinical drug trials because they may not reflect real world outcomes.

## Quality assessment

In our prior review, we developed quality criteria relevant to studies of unfavorable outcomes in India's TB care cascade [5], because of variable guidelines for assessing the quality of these observational studies and the variability in study approaches across gaps. For example, Gap 1 studies involved population-based screening to identify people with TB symptoms (with risks of bias including suboptimal sampling and survey response rates); Gaps 2 and 3 involved follow-up of people not yet engaged in TB care (with risks including suboptimal follow-up approaches); while Gap 4 and 5 studies involved cohort or case-control designs. For the current review, we used quality criteria similar to those in the prior review, because these criteria describe methodological rigor and risk of bias (Table B in each of the S1–S5 Appendices). Studies were only excluded based on quality if they used convenience sampling, which risks being nonrepresentative.

## Data extraction

Two reviewers (from TJ, DJ, AG, or DV) independently extracted data on study design, location, sample size, exposures, and outcomes into a structured Excel spreadsheet. Disagreements were resolved by a third reviewer (RS). For studies reporting factors associated with unfavorable outcomes, we extracted unadjusted and adjusted effect estimates and 95% confidence intervals (CIs) for all exposures. For studies that did not report effect estimates, when possible, we used the findings to estimate unadjusted odds ratios with 95% CIs. For studies reporting reasons for unfavorable outcomes, we extracted the proportion of individuals reporting each reason. When 95% CIs were not reported, we estimated these using the binomial "exact" method, assuming an infinite population size [20].

We reported unadjusted and adjusted effect estimates, including 95% CIs and *p*-values as reported in the original papers, in Table D in each of the S1–S5 Appendices. In the Forest plots and main text, we only presented statistically significant adjusted effect estimates, as these may represent associations from higher-quality studies. For exposure variables with statistically significant findings, we also reported any nonsignificant adjusted effect estimates from included studies for these same variables in the main text and Forest plot captions. Dependent variables that were adjusted for can be found in Table D in each of the S1–S5 Appendices. Some studies presented the association of exposures with favorable (rather than unfavorable) outcomes. For consistency, we "flipped" these effect estimates to present associations with unfavorable outcomes. For some variables, we changed the reference group for consistency. For example, because most studies compared men to the reference group of women, we "flipped" effect estimates for studies presenting men as the reference group.

## Framework for organizing and visualizing findings

Informed by the social-ecological model [21,22]—which looks at the interplay of risk factors at the individual, family, and society levels—we organized findings into "person-, family-, or society-related factors" and "health system factors," with subcategories, to inform future interventions (Table 2). To visualize common or discordant findings, we generated Forest plots of statistically significant adjusted effect estimates, organized by framework subcategories, using Stata version 16.1 (College Station, USA). Meta-analyses were not performed because of the diverse exposures and heterogeneity of findings. We also generated Forest plots of proportions for reasons reported by people with TB for unfavorable outcomes.

To visualize trends, we combined different unfavorable outcomes (e.g., death, loss to follow-up) or effect estimates (i.e., odds, relative risk, hazard ratios) in the same Forest plots while denoting this in footnotes. We occasionally modified language in the original studies

**Table 2. Framework for organizing and visualizing barriers contributing to unfavorable TB care cascade outcomes.**

| Subcategories for organizing review findings | Examples of factors included in each subcategory[a] |
|---|---|
| **Person-, family-, and society-related factors** | |
| Demographic factors | Age, sex, marital status, religion, urban or rural residence, region of origin, caste |
| Socioeconomic factors | Income, literacy, educational attainment, employment, working hours, rural or urban setting |
| Patient mobility | Work migration, travel for other reasons, inability of healthcare providers to find people because they moved from their address |
| TB-related clinical factors | Prior TB history, drug susceptibility or resistance, site of TB (e.g., pulmonary versus extrapulmonary), disease severity, medication adverse effects |
| Other clinical factors | Nutritional status, HIV, diabetes, structural lung disease |
| Substance use | Alcohol use, tobacco use, injection drug use |
| Psychological factors | Depression, anxiety, medication nonadherence, internalized stigma |
| Knowledge-related factors | TB knowledge, health system knowledge |
| Family-related factors | Social support, accompaniment to clinic visits, enacted stigma within the family |
| Society-related factors | Enacted stigma or discrimination, social activities or holiday festivities delaying care |
| **Health system factors** | |
| Perceptions of the health system | Distrust of health services, dissatisfaction with health services, care-seeking at multiple sites |
| Healthcare accessibility (i.e., structural barriers) | Logistical and geographical accessibility (e.g., distance to clinic, waiting times), financial accessibility (e.g., cost of travel, cost of care) |
| Navigational challenges | Difficulties in navigating within or between facilities (i.e., understanding when, where, or how to get to care) |
| Infrastructural limitations | Electricity failures, electronic health record failures, or bed shortages |
| Health sector- or facility-related factors | Type of health sector (public or private) or facility (primary, tertiary, etc.) |
| Healthcare provider factors | Absenteeism, negative interactions of healthcare providers with people with TB, understaffing of facilities, stigmatization of people with TB by healthcare workers |
| Approaches to care provision | Monitoring approach used, challenges engaging with directly observed therapy, use of cellphone-based reminders, cash transfers |
| Quality of care | Not identifying people with presumptive TB, poor recording of contact information of people with TB, not providing adequate counseling, TB drug stockouts |

[a] The factors included in this column are meant to be representative of the types of factors that might be included in each subcategory; they do not represent an exhaustive list of all possible factors within each subcategory.

HIV, human immunodeficiency virus; TB, tuberculosis.

(without changing the meaning) to enable visualization on Forest plots or to ensure use of person-centered language [23]. We changed older TB terminology (e.g., category 2 treatment) to reflect contemporary terminology (e.g., person with previously treated TB).

## Results

### Characteristics and quality of the included studies

Table 3 summarizes characteristics and quality of studies that had findings reported in the main manuscript. Studies describing findings from unadjusted (i.e., univariate) regression

**Table 3. Characteristics and quality of the studies reported in the main manuscript for each TB care cascade gap.**

| Study characteristics | Gap 1 | Gap 2 | Gap 3 | Gap 4 | Gap 5 |
|---|---|---|---|---|---|
| **Studies contributing to findings reported in the main manuscript** | | | | | |
| Total studies contributing to findings in the main manuscript (i.e., reporting adjusted "factors" or "reasons" for unfavorable outcomes) | 14 | 17 | 15 | 86 | 14 |
| Studies reporting factors associated with unfavorable outcomes from adjusted (i.e., multivariable) regression analyses | 7[a] | 9[b] | 5[d] | 71[f] | 14[h] |
| Studies reporting reasons from surveys of people who experienced unfavorable outcomes | 9 | 10[c] | 11[e] | 18[g] | 0 |
| **Study setting** | | | | | |
| States or union territories covered by included studies | 7 | 8 | 8 | 24 | 6 |
| Studies conducted in a rural setting only | 4 | 2 | 2 | 16 | 4 |
| Studies conducted in an urban setting only | 8 | 6 | 4 | 45 | 5 |
| Studies conducted in both rural and urban settings | 3 | 9 | 9 | 28 | 5 |
| **Quality or risk of bias criteria** | | | | | |
| Studies that were medium or low quality for sampling strategy | 0 | 0 | 0 | 0 | 0 |
| Studies that were medium or low quality for sample size | 2 | 0 | 2 | 17 | 0 |
| Studies that were medium or low quality for the proportion of the estimated population screened for TB symptoms (Gap 1 only) | 9 | NA | NA | NA | NA |
| Studies that were medium or low quality for the proportion of individuals with TB symptoms who were interviewed (Gap 1 only) | 4 | NA | NA | NA | NA |
| Studies that were medium or low quality for the time frame of research fieldwork after start of diagnostic evaluation (Gaps 2 and 3 only) | NA | 10 | 11 | NA | NA |
| Studies that were medium or low quality for only using outcomes self-reported by the government TB program (Gaps 2 and 3 only) | NA | 4 | 3 | NA | NA |
| Studies that were medium or low quality for retrospective assessment of both exposures and outcomes (Gap 4 only) | NA | NA | NA | 29 | NA |
| Studies that were medium or low quality for only conducting passive surveillance for posttreatment TB recurrence or mortality (Gap 5 only) | NA | NA | NA | NA | 6 |
| Studies that were medium or low quality for only diagnosing TB recurrence clinically or only testing people with symptoms (Gap 5 only) | NA | NA | NA | NA | 6 |

[a]Of these 7 studies, 1 study reported data from 2 locations, so we present study characteristics across 8 locations.

[b]Of these 9 studies, 3 studies reported on non-pursual of the diagnostic workup despite referral; 1 study reported on non-completion of sputum microscopy; 2 studies reported on chest X-ray non-completion; and 3 studies reported on NAAT, line probe assay, or culture non-completion.

[c]Of these 10 studies, 2 studies reported on non-completion of sputum microscopy evaluation; 2 studies reported on chest X-ray non-completion; and 6 studies reported on NAAT non-completion.

[d]Of these 5 studies of pretreatment loss to follow-up, 4 studies evaluated people with drug-susceptible TB, and 1 study evaluated people with drug-resistant TB.

[e]Of these 11 studies of pretreatment loss to follow-up, 9 studies evaluated people with drug-susceptible TB, and 2 studies evaluated people with drug-resistant TB.

[f]Of these 71 studies, 51 studies evaluated people with drug-susceptible TB; 14 studies evaluated people with drug-resistant TB; 5 studies evaluated people with HIV being treated for TB; and 1 study evaluated children with TB.

[g]Of these 18 studies, 15 studies evaluated people with drug-susceptible TB, and 3 studies evaluated people with drug-resistant TB.

[h]Of these 14 studies, 10 studies reported findings on TB recurrence as a single outcome or part of a composite outcome, and 4 studies reported findings on posttreatment mortality as a single outcome or part of a composite outcome with on-treatment mortality.

HIV, human immunodeficiency virus; NA, not applicable (i.e., quality indicator not relevant to a specific gap); NAAT, nucleic acid amplification test; TB, tuberculosis.

analyses alone are not reported in the main manuscript; however, these findings are in Table D in each of the S1–S5 Appendices. Summaries of the characteristics and quality of all studies, including those only reporting findings from unadjusted analyses, are in the S6 Appendix (Table A and the narrative text).

## Gap 1—Barriers contributing to people with TB symptoms not having sought care

**Factors associated with not having sought care for TB symptoms in regression analyses.** Across 7 studies reporting adjusted analyses, men had higher odds of not seeking care in

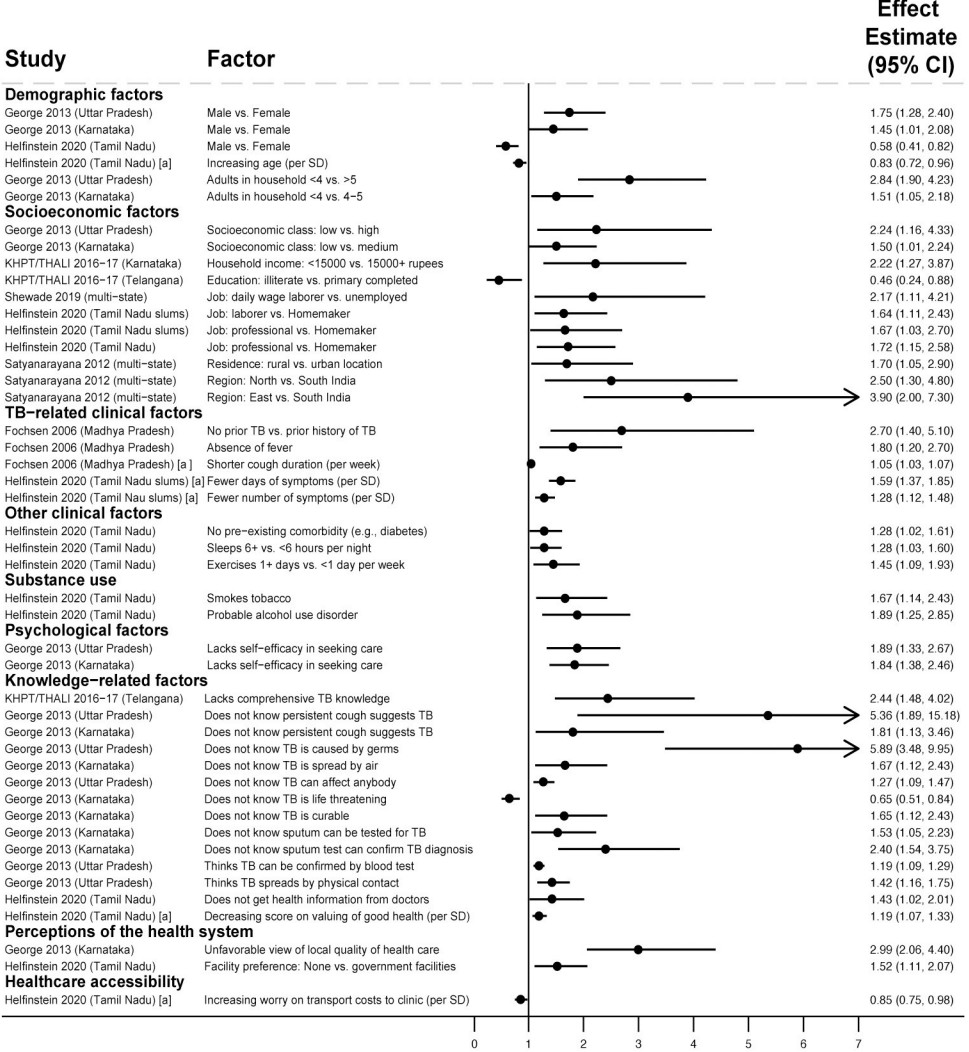

**Fig 1. Factors associated with people in the community not having sought care for TB symptoms (Gap 1).** All studies used multivariable logistic regression with findings reported as adjusted odds ratios [24–30]. Estimates greater than 1 represent increased odds of not seeking care; estimates less than 1 represent decreased odds of not seeking care. Arrowheads mean that the upper limit of the CI extends beyond the end of the x-axis. Variables labeled [a] represent continuous variables in regression analyses; effect estimates should be interpreted per one level change in the unit in parentheses. Only statistically significant findings are presented. Some studies in the review with adjusted analyses reported nonsignificant findings for sex [26–29], number of adults in the household [29], age [24,26–29], income or socioeconomic class [24,25], educational attainment [24,25,27–29], employment status [24,25,29], and TB knowledge [27]. CI, confidence interval; KHPT/THALI, Karnataka Health Promotion Trust/Tuberculosis Health Action Learning Initiative; SD, standard deviation; TB, tuberculosis.

2 states in 1 study [24] (Fig 1 and Table D in the S1 Appendix). Women had higher odds of not seeking care in 1 study, but only after adjusting for smoking and alcohol use, which were reported exclusively by men and were also associated with not seeking care [25]. Sex had non-significant associations in 4 studies [26–29]. Lower socioeconomic status—measured using income [24,26,27], daily wage labor [25,29], or living in rural locations or lower-income states [28]—was associated with higher odds of not seeking care; lower socioeconomic status had nonsignificant associations in 5 studies [24,25,27–29] and an inverse association for educational status in 1 study [26].

Not having TB previously and lower symptom severity—e.g., absence of fever, fewer symptoms, and shorter duration [25,30]—were associated with higher odds of not seeking care. Lacking knowledge of symptoms, transmission, and the curability of TB were associated with higher odds of not seeking care [24–26]; TB knowledge had a nonsignificant association in 1 study [27]. Viewing local clinic quality unfavorably [24] and lacking preference for government clinics [25] were associated with higher odds of not seeking care.

**Reasons for people with TB symptoms not seeking care.**   Across 9 studies reporting reasons for not seeking care, financial and work constraints were reported frequently, with 3 studies finding about half of people (44% [83/187] [31], 48% [968/2,016] [32], and 51% [194/381] [24]) faced these barriers (Fig 2). Lack of symptom severity or resolving symptoms were reported commonly, with these barriers experienced by 26% (42/162) [33] to 63% (35/56) [34] of people across 7 studies [24,31,32,34–37]. In specific studies, 19% (74/381) reported dissatisfaction with local clinics [24], 29% (589/2,016) reported healthcare provider indifference [32], while 30% (603/2,016) reported distance and 26% (8/31) reported lack of transport prevented them from seeking care [32,36].

## Gap 2—Barriers to completing the TB diagnostic workup

**Factors associated with not pursuing diagnostic workup despite referral in regression analyses.**   Across 3 studies reporting adjusted analyses, not having a prior TB history [38], lower symptom severity [38,39], alcohol use [39], and missing data on age or human immunodeficiency virus (HIV) status in medical records [39,40] were associated with higher risk of non-pursual of workup (Fig A in the S6 Appendix and Table D in the S2 Appendix). Alcohol use had a nonsignificant association in 1 study [38]. People referred by community health workers (e.g., Anganwadi workers) had higher risk of workup non-pursual compared to those referred by registered medical providers (i.e., physicians) [38]. People referred by government TB units or private clinics had higher risk of workup non-pursual compared with those referred by peripheral health institutes (i.e., primary health centers) [40].

**Factors associated with non-completion of sputum microscopy evaluation in regression analyses.**   In 1 study reporting an adjusted analysis, older age (>50 years), shorter symptom duration (< = 15 days), lack of an accompanying person for clinic visits, and not being informed by providers about concern for TB were associated with higher odds of sputum microscopy non-completion [41] (Fig B in the S6 Appendix and Table D in the S2 Appendix).

**Reasons for non-completion of sputum microscopy evaluation.**   In 2 studies, reasons for non-completion of sputum microscopy included work constraints (reported by 15% [14/92] in 1 study [42]), symptom improvement (reported by 22% [70/314] in 1 study [41]), and health system barriers, especially negative interactions with, or unavailability of, providers (reported by 45% [144/314] in 1 study [41]) (Fig C in the S6 Appendix).

**Factors associated with chest X-ray non-completion in regression analyses.**   In 2 studies reporting adjusted analyses, not being able to afford a private sector X-ray [43], being below the poverty line and >30 kilometers from a public X-ray facility [43], evaluation at a district (versus subdistrict) hospital [44], and not being informed by providers that an X-ray was needed [43] were associated with chest X-ray non-completion (Fig D in the S6 Appendix and Table D in the S2 Appendix).

**Reasons for chest X-ray non-completion.**   In 2 studies, reasons for chest X-ray non-completion included work constraints (reported by 25% (16/65) in 1 study [45]), symptom improvement (reported by 67% [162/243] in 1 study [46]), and not being informed about the need for further workup (reported by 25% [16/65] in 1 study [45]) (Fig E in the S6 Appendix).

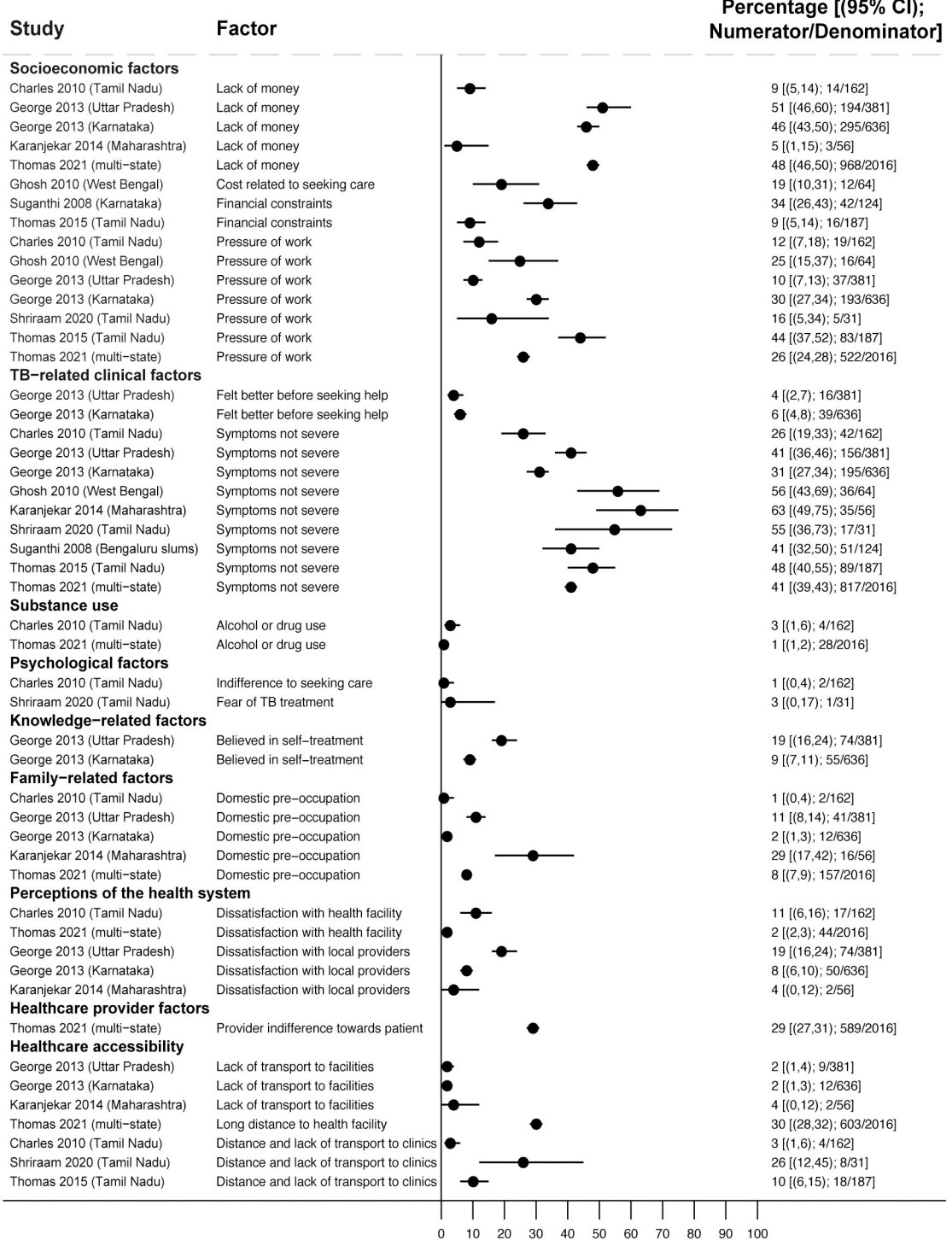

**Fig 2. Reasons why people with TB symptoms in the community had not sought care (Gap 1).** All studies report the percentage of people reporting a given reason for not seeking care [24,31–37]. The denominator comprises people who had not sought care in population-based surveys. CI, confidence interval; TB, tuberculosis.

**Factors associated with nucleic acid amplification testing (NAAT) or mycobacterial culture non-completion in regression analyses.** Across 3 studies reporting adjusted analyses, people >64 years [47], with extrapulmonary or smear–negative pulmonary disease (as compared to smear–positive pulmonary disease [47–49]), with an indication for NAAT other than treatment failure [47], and who were evaluated at medical colleges [47] had higher NAAT non-completion (Fig F in the S6 Appendix and Table D in the S2 Appendix). Extrapulmonary TB had a nonsignificant association in 1 study [48].

**Reasons for NAAT or mycobacterial culture non-completion.** Across 6 studies reporting reasons for NAAT or culture non-completion, providers missed identifying 8% (47/628) [50] to 54% (417/770) [47] of people with drug-resistant TB risk factors who merited NAAT (5 studies [47,50–53]) or culture (1 study [54]) (Fig G in the S6 Appendix). Loss of sputum samples during transfer to reference laboratories occurred for 3% (17/628) [50] to 32% (108/341) [52] of people across 5 studies [47,50,52–54].

## Gap 3—Barriers contributing to pretreatment loss to follow-up (PTLFU)

**Factors associated with PTLFU among people with drug-susceptible TB in regression analyses.** Across 4 studies reporting adjusted analyses, male sex [55], older age (>30 [56] or >50 years [57]), being in the lowest wealth tertile [56], lack of secondary education [56], long distance to clinic (i.e., living in a rural area but seeking evaluation in a city [57]), having a previous TB treatment history [40,57], and tobacco use [55] were associated with PTLFU (Fig 3 and Table D in the S3 Appendix). Sex had nonsignificant associations in 3 studies [40,56,57], age in 2 studies [40,55], educational attainment in 1 study [55], and previous TB treatment history in 1 study [55].

Regarding health system factors, people whose families preferred private sector services [56], who were evaluated at private sector clinics or labs [40], or who were evaluated at high-volume diagnostic facilities [57] had higher risk of PTLFU. People whose contact information in the diagnostic register was missing or unreadable (making them untrackable) had higher PTLFU [57]. Concordant with this finding, phone calls to notify people of their TB diagnosis or remind them to start treatment were associated with lower PTLFU [55].

**Reasons for PTLFU in people with drug-susceptible TB.** Across 9 studies of people who experienced PTLFU, reasons included work constraints (reported by 5% [1/20] [58] to 32% [42/132] [59] in 2 studies); mobility and work-related migration (reported by 9% [218/2,494] [60] to 31% [41/132] [59] in 3 studies [59–61]); psychological barriers such as stigma, disbelief in the diagnosis, or treatment refusal (reported by 5% [28/552] [61] to 25% [627/2,494] [60] in 5 studies [41,58,60–62]); concerns about adverse effects during prior TB treatment (reported by 42% [41/98] in 1 study [62]); and alcohol use (reported by 20% [4/20] in 1 study [58]) (Fig 4). Of people experiencing PTLFU, 4% (4/98) [62] to 40% (1,007/2,494) [60] died before starting treatment in 5 studies [57,60–63].

Health system barriers contributing to PTLFU included distrust or negative perceptions of government services (reported by 2% [3/132] [59] to 45% [44/98] [62] in 3 studies [41,59,62]); drug stockouts (reported by 2% [3/132] [59] to 14% [6/43] [41] in 2 studies); long travel distance to clinic (reported by 33% [7/21] in 1 study [64]); inability to engage with clinic-based directly observed therapy (DOT; reported by 14% [6/43] in 1 study [41]); and inability of providers to contact people due to poor recording of contact information (affecting 21% [525/2,494] [60] to 52% [285/552] [61] in 4 studies [57,60,61,63]).

**Factors associated with PTLFU in people with drug-resistant TB in regression analyses.** In 1 study reporting an adjusted analysis, people whose indication for drug susceptibility testing was TB treatment failure (versus TB recurrence) had a relative risk of PTLFU of 6.0

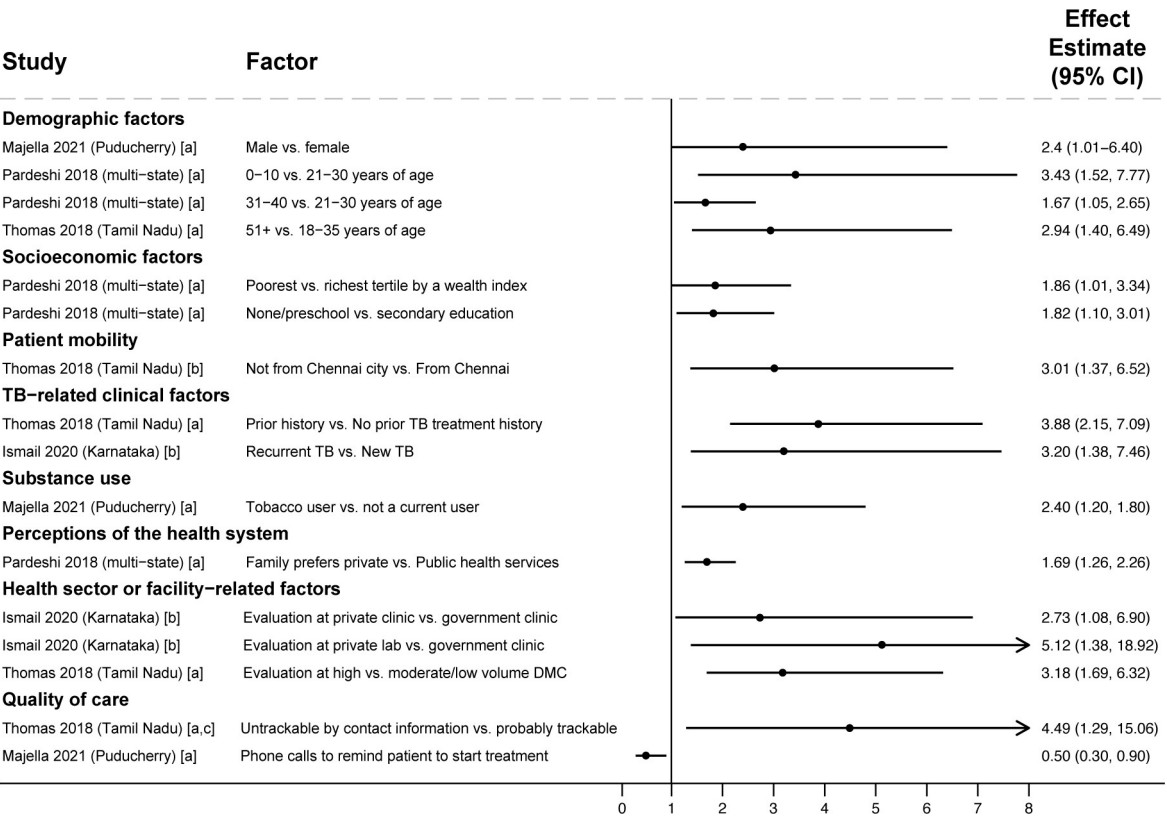

**Fig 3. Factors associated with PTLFU after diagnosis among people with drug-susceptible TB (Gap 3).** All studies [55–57] used multivariable regression with findings reported as adjusted odds ratios, except Ismail 2020 [40], which reported findings as adjusted risk ratios. Estimates greater than 1 represent increased risk of PTLFU; estimates less than 1 represent decreased risk of PTLFU. Study labels indicate: [a] outcome was non-registration in the TB program, [b] outcome was not starting TB treatment, and [c] inability to track people with TB due to poor recording of phone or address information in diagnostic registers. Only statistically significant findings are presented. Some studies with adjusted analyses reported nonsignificant associations for sex [40,56,57], age [40,55], educational attainment [55], and previous TB treatment [55]. CI, confidence interval; DMC, designated microscopy center; PTLFU, pretreatment loss to follow-up; TB, tuberculosis.

(95% CI 2.3, 15.2) [65] (Table D in the S3 Appendix). People whose sputum microscopy result was missing, suggesting incomplete clinical evaluation, had a relative risk of PTLFU of 17.1 (95% CI 7.7, 39.3) compared to people with a positive sputum result.

**Reasons for PTLFU in people with drug-resistant TB.** In 2 studies of people with drug-resistant TB who experienced PTLFU, reasons included inability of providers to track people due to poor recording of contact information (affecting 21% [10/48] in 1 study [54]), death before starting treatment (affected 17% [1/6] [51] to 35% [17/48] [54] in 2 studies), and treatment refusal (reported by 83% [5/6] in 1 study [51]) (Fig H in the S6 Appendix).

## Gap 4—Barriers to treatment success among people who start TB treatment

We divide Gap 4 findings into subpopulations of people with: (1) drug-susceptible TB; (2) drug-resistant TB (i.e., isoniazid monoresistant, rifampin-resistant [RR], and multidrug-resistant [MDR] TB); (3) TB in people with HIV; and (4) TB in children. Given extensive findings, we report results for people with drug-susceptible TB in the following subsections: demographic factors; clinical barriers; socioeconomic, psychosocial, and family- or society-related barriers; and health system barriers.

**Demographic factors contributing to unfavorable treatment outcomes among people with drug-susceptible TB.** Across 51 studies reporting adjusted analyses including people with drug-susceptible TB, male sex and older age were associated with unfavorable treatment outcomes in 15 [41,66–79] and 15 [68,69,72,75–86] studies, respectively (Fig I in the S6 Appendix and Table D in the S4 Appendix). Sex and age had nonsignificant associations in 21 [67,68,72,77,80,82,84–98] and 18 [41,68,71,72,76–78,83,87,89,91–94,98–101] studies, respectively. While being married was associated with unfavorable treatment outcomes when compared to having never been married in 2 studies [41,73], people who were married had better outcomes when compared to those who were separated or divorced in 1 study [100]. Marital status had nonsignificant associations in 4 studies [72,73,92,94].

**Clinical barriers contributing to unfavorable treatment outcomes among people with drug-susceptible TB.** Across 51 studies reporting adjusted analyses including people with drug-susceptible TB, indicators of diagnostic delay or advanced TB—including illness >2 months in 1 study [102], advanced radiographic (e.g., cavitary) disease in 2 studies [102,103], and smear–positive or microbiologically diagnosed TB (versus smear–negative, extrapulmonary, or clinically diagnosed TB) in 11 studies [68,70,71,73,76,81,85,87,89,97,104]—were associated with unfavorable treatment outcomes (Fig J in the S6 Appendix and Table D in the S4 Appendix). People with more severe symptoms at treatment initiation were less likely to experience loss to follow-up in 2 studies [91,105], but more likely to experience composite unfavorable TB treatment outcomes in 1 study [99]. There were nonsignificant associations for illness duration in 1 study [91], smear–positive pulmonary disease in 11 studies [66,68–70,72,75,82,88,95–97], and number of symptoms at treatment initiation in 2 studies [86,91].

Among studies reporting adjusted analyses including both new and previously treated individuals, previously treated individuals had poorer treatment outcomes in 11 analyses [66,67,69,71–73,84,88,91,97,104]; previous TB treatment history had nonsignificant associations in 16 analyses [68,78,80,82–84,87,90,91,93–99]. Among studies with adjusted analyses that only included previously treated individuals, the outcome of a person's prior treatment predicted subsequent outcomes. Compared to people who completed their prior TB treatment —and who therefore had disease recurrence—people who were previously lost to follow-up [78,91,96] or experienced treatment failure [79,96] were more likely to have unfavorable treatment outcomes.

Among studies including people with drug-susceptible and drug-resistant TB, drug resistance (including isoniazid monoresistance) was associated with unfavorable treatment outcomes in 2 studies [72,79]; drug resistance had a nonsignificant association in 1 study [106]. Medication-related issues—subtherapeutic rifampin levels in 1 study [107] and TB drug adverse effects in 3 studies [108–110]—were also associated with unfavorable treatment outcomes; adverse effects had nonsignificant associations in 2 studies [98,105].

People with HIV, or unknown HIV status, had higher unfavorable outcomes in 10 studies [69,71–73,76,77,79,82,110] (Fig K in the S6 Appendix and Table D in the S4 Appendix); HIV status had nonsignificant associations in 7 studies [68,73,77,79,87,93,106]. In 5 studies, pre-treatment undernutrition—assessed by weight, body mass index (BMI), or stunting [72,77,82,84,96]—or non-improvement in nutritional status with treatment [77] were associated with unfavorable outcomes. Low BMI had a nonsignificant association in 1 study [98]. People with untreated diabetes or unknown diabetes status (versus not having diabetes) had higher unfavorable outcomes in 2 studies [69,111]. However, diabetes was protective in 2 studies [77,100] and had non-significant associations in 4 studies [68,72,93,98].

Across 15 studies surveying people who experienced loss to follow-up or medication non-adherence during treatment, clinical reasons included medication side effects (reported by 7% [1/14] [112] to 47% [15/32] [92] of people across 15 studies [87,92,102,106,112–122]), long

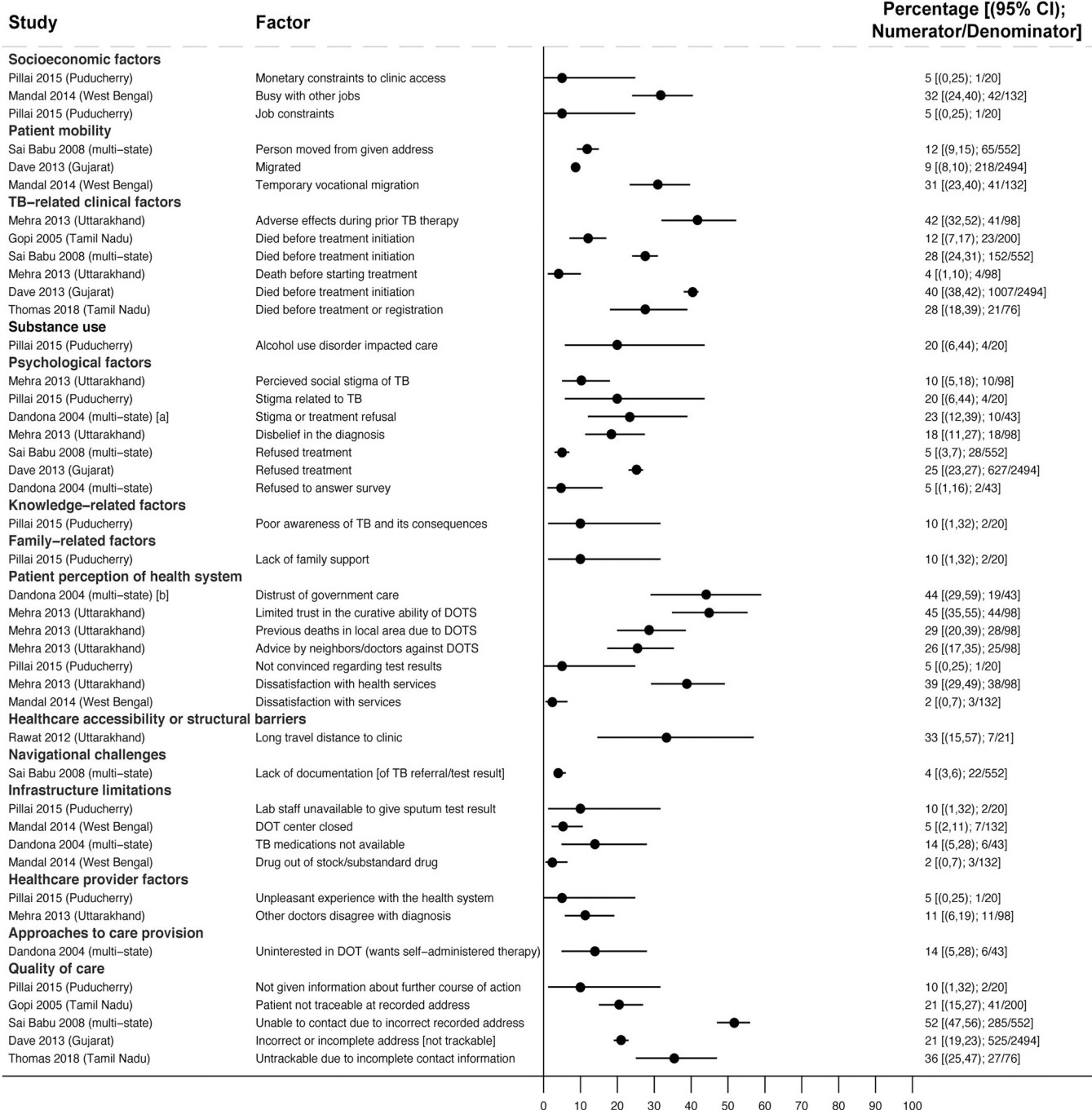

**Fig 4. Reasons for PTLFU among people with drug-susceptible TB (Gap 3).** All studies [41,57–64] report estimates of the percentage of people interviewed who reported a given reason for not starting on, or registering in, TB treatment. Study labels indicate: [a] summarizes the following responses: "no time or busy, afraid that someone would come to know of disease, was very sick, and did not know about TB treatment;" [b] summarizes the following: "did not want treatment at a government center, did not have belief in government doctors, and unable to meet the doctor." CI, confidence interval; DOT, directly observed therapy; PTLFU, pretreatment loss to follow-up; TB, tuberculosis.

treatment duration (reported by 11% [8/70] [121] to 16% [5/32] [92] of people in 2 studies), early symptom improvement (reported by 4% [1/28] [119] to 55% [110/201] [114] across 11 studies [87,92,106,113–117,119–121]), and lack of symptom improvement (reported by 1% [1/

150] [117] to 34% [11/32] [92] across 8 studies [92,113–115,117,120–122]) (Fig L in the S6 Appendix).

**Socioeconomic, psychosocial, and family- or society-related barriers contributing to unfavorable treatment outcomes in people with drug-susceptible TB.** Across 51 studies reporting adjusted analyses including people with drug-susceptible TB, people who were illiterate or had fewer years of education (in 5 studies [72,90,91,104,109]), were living in kaccha (informal) homes (in 1 study [99]), or were living in homes with indoor air pollution (in 1 study [89]) were more likely to have unfavorable treatment outcomes (Fig 5). Educational attainment had nonsignificant associations in 9 studies [41,66,84–86,91,92,94,108]. Being a daily wage laborer (which indicates lower socioeconomic status) was associated with unfavorable outcomes in 1 study [87]; however, being employed (versus being unemployed) was associated with unfavorable outcomes in 2 studies [67,90]. Employment status had nonsignificant associations in 5 studies [84,85,87,93,94].

Inadequate TB knowledge was associated with unfavorable outcomes in 4 studies [93,105,109,110]; TB knowledge had nonsignificant associations in 2 studies [91,123]. Lack of family support or supervision was associated with unfavorable outcomes in 2 studies [99,110]. In 2 studies, discrimination due to TB was associated with unfavorable outcomes [66,89]. Family support and stigma or discrimination had nonsignificant associations in 4 studies [73,93,105,123] and 2 studies [93,101], respectively.

Current or past history of smoking was associated unfavorable outcomes in 5 studies [71,77,92,105,123]; smoking had nonsignificant associations in 9 studies [66,84,86,93,98,108,109,123]. Alcohol use was associated with unfavorable outcomes in 13

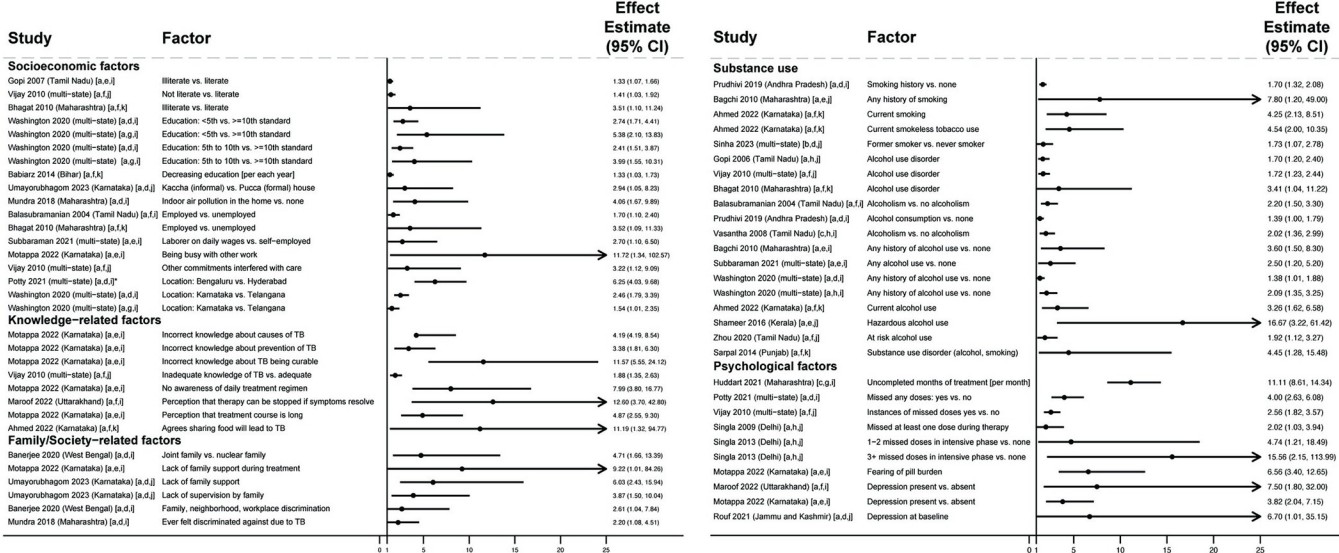

**Fig 5. Socioeconomic, psychosocial, and family- or society-related factors associated unfavorable treatment outcomes in people with drug-susceptible TB (Gap 4).** All studies used multivariable regression and reported adjusted effect estimates [66,67,71–73,77,81,84,86,87,89–94,99–105,108–110,123]. Estimates greater than 1 represent increased risk of unfavorable outcomes; estimates less than 1 represent decreased risk of unfavorable outcomes. Arrowheads mean the upper limits of the CI extend beyond the end of the x-axis. Study labels indicate effect estimates are: [a] odds ratios; [b] incidence rate ratios; or [c] hazard ratios. Study labels indicate outcomes are: [d] any unfavorable treatment outcome; [e] medication nonadherence; [f] loss to follow-up; [g] death; or [h] treatment failure. Study labels indicate that participants are: [i] from a combined population of people with new TB or a prior TB treatment history; [j] people with new TB only; or [k] people with a prior TB treatment history only. Only statistically significant findings are presented. Some studies in the review with adjusted analyses reported nonsignificant findings for educational attainment [41,66,84–86,91,92,94,108], employment status [84,85,87,93,94], TB knowledge [91,123], family support [73,93,105,123], stigma [93,101], smoking [66,84,86,93,98,108,109,123], alcohol use [77,85,93,98,106,124], and medication nonadherence [106]. CI, confidence interval; TB, tuberculosis.

studies [67,71,72,84,86,87,90,92,100,105,108,109,123]; alcohol use had nonsignificant associations in 6 studies [77,85,93,98,106,124]. In 5 studies, medication nonadherence was associated with unfavorable outcomes [73,95,102,103,109]; nonadherence had a nonsignificant association in 1 study [106]. In 4 studies, depression was associated with unfavorable outcomes [93,94,101,110].

Across 15 studies surveying people who experienced loss to follow-up or medication nonadherence, reported reasons included: work constraints (reported by 1% [1/82] [106] to 38% [12/32] [92] of people across 8 studies [92,106,113–115,118–120]); migration or travel (reported by 1% [2/201] [114] to 91% [20/22] [103] of people across 9 studies [87,103,106,113,114,116,118,120,121]); lack of knowledge of treatment duration or of the risks of treatment interruption (reported by 7% [5/70] [121] to 25% [14/55] [122] of people in 4 studies [114,115,121,122]); TB stigma (reported by 3% [4/141] [113] to 81% [26/32] [92] of people across 3 studies [87,92,113]); alcohol use (reported by 3% [5/150] [117] to 35% [29/82] [106] of people across 5 studies [92,106,113,114,117]); forgetfulness in dose-taking (reported by 19% [15/78] [118] to 43% [6/14] [112] of people across 4 studies [87,112,118,122]); and depression (reported by 7% [27/377] [112] to 23% [39/167] [87] in 2 studies) (Fig 6).

**Health system barriers contributing to unfavorable treatment outcomes in people with drug-susceptible TB.** Across 51 studies reporting adjusted analyses including people with drug-susceptible TB, dissatisfaction with TB services [41,89,105,109,110], and negative interactions [105,109,110] or lack of support [99] from healthcare providers were associated with unfavorable outcomes in 6 studies (Fig 7 and Table D in the S4 Appendix). Dissatisfaction with TB services had a nonsignificant association in 1 study [93]. Barriers to healthcare accessibility—including living a long distance from clinics, spending greater time collecting medications, and paying for medications or clinic transportation—were associated with unfavorable outcomes in 6 studies [87,91,104,105,123,125]. Distance to the nearest TB clinic and paying for treatment had nonsignificant associations in 4 studies [41,91,93,108].

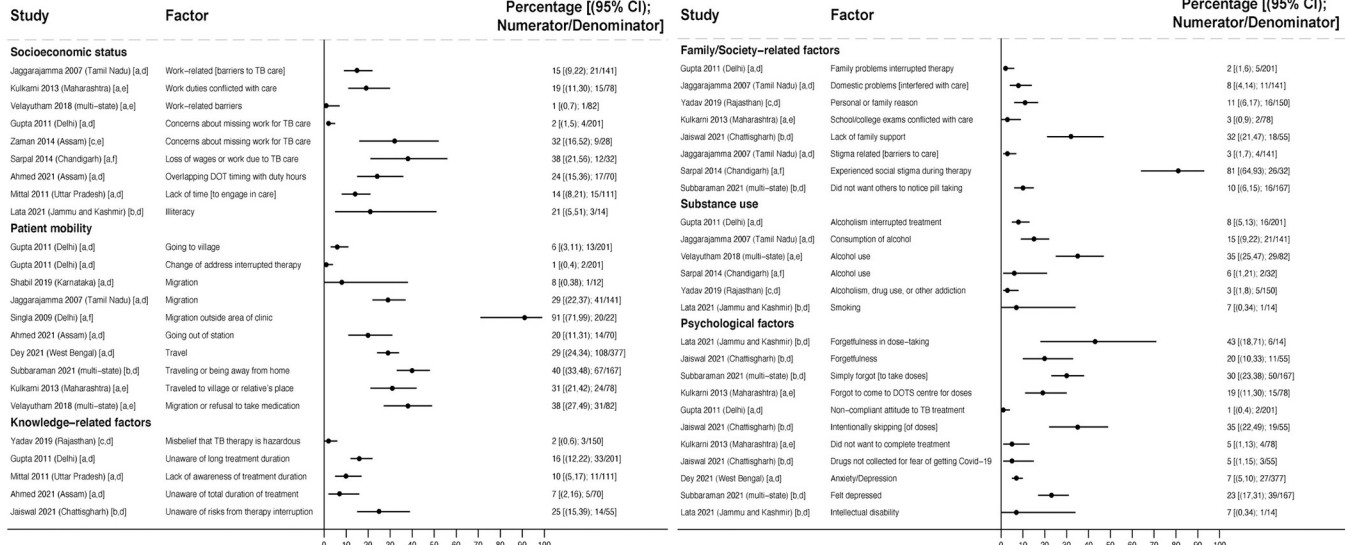

**Fig 6. Socioeconomic, psychosocial, and family- or society-related reasons reported for experiencing unfavorable treatment outcomes in people with drug-susceptible TB (Gap 4).** All studies reported estimates of the percentage of people who reported a given reason for experiencing unfavorable outcomes [87,92,103,106,112–122]. Study labels indicate outcomes are: [a] LTFU, [b] medication nonadherence, or [c] any interruption (defined as a combination of medication nonadherence and LTFU). Study labels indicate patient populations are: [d] from a combined population of people with new TB or a prior TB treatment history; [e] people with new TB only; [f] people with a prior TB treatment history only. CI, confidence interval; DOTS, directly observed therapy short course; LTFU, loss to follow-up; TB, tuberculosis.

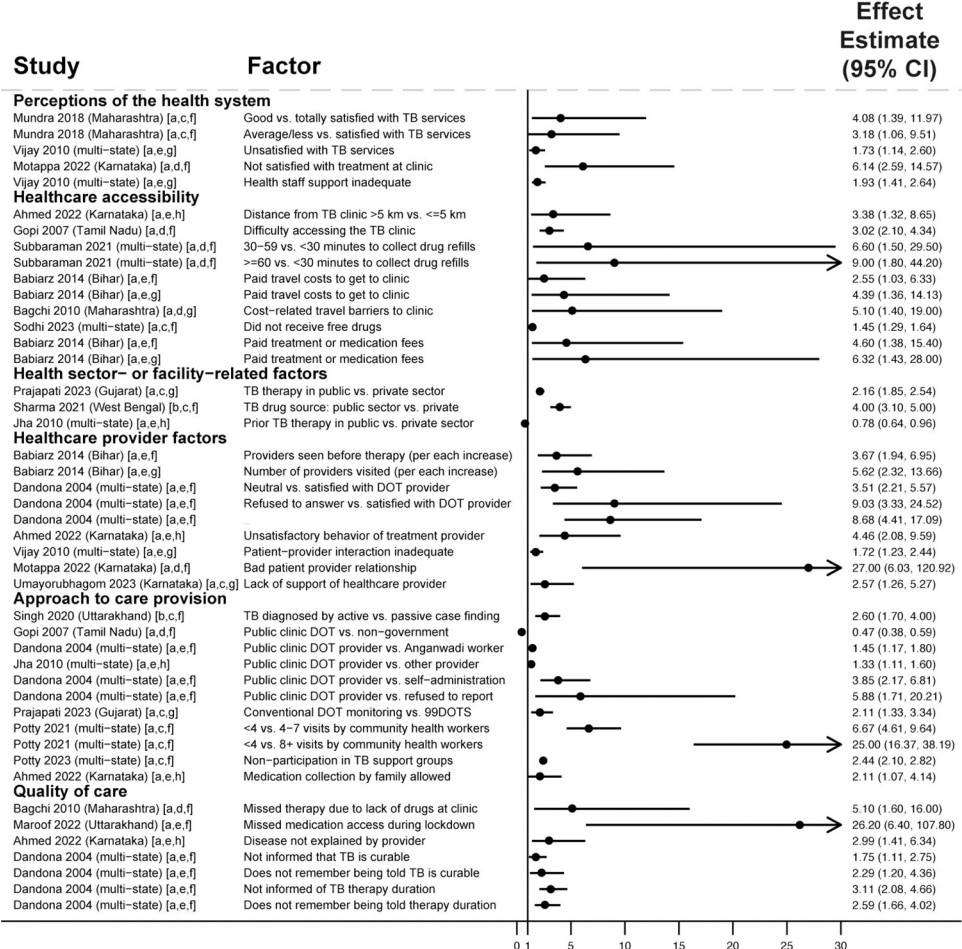

**Fig 7. Health system factors associated with unfavorable treatment outcomes in people with drug-susceptible TB (Gap 4).** All studies used multivariable regression and reported adjusted effect estimates [41,73,76,78,83,87–89,91,93,104,105,109,110,123,125]. Estimates greater than 1 represent increased risk of unfavorable outcomes; estimates less than 1 represent decreased risk of unfavorable outcomes. Arrowheads mean that the upper limits of the CI extend beyond the end of the x-axis. Study labels indicate effect estimates are: [a] odds ratios; or [b] relative risk ratios. Study labels indicate outcomes are: [c] any unfavorable treatment outcome; [d] medication nonadherence; or [e] loss to follow-up. Study labels indicate that participants are: [f] from a combined population of people with new TB or a prior TB treatment history; [g] people with new TB only; or [h] people with a prior TB treatment history only. Only statistically significant findings are presented. Some studies in the review with adjusted analyses reported nonsignificant findings for health system dissatisfaction [93], proximity to the nearest clinic [41,93,108], treatment costs [91,93], number of providers visited [91], type of DOT provider [41,84], type of case finding approach [127], type of adherence monitoring approach [75,84,105,109], and incorrect and/or inadequate information given by providers [73,93,123]. Anganwadi workers are government-supported community health workers. 99DOTS is a TB digital adherence technology. CI, confidence interval; DOT, directly observed therapy; DOTS, directly observed therapy short course; km, kilometer; TB, tuberculosis.

Outcomes were also influenced by the health sector or approaches to care provision. Receiving care in the public (versus the private) sector was associated with unfavorable outcomes in 2 studies [76,83]; however, for people with previously treated TB, prior private sector treatment was associated with unfavorable outcomes [78]. With the exception of 1 study [104], DOT by a public sector provider (usually requiring people to visit clinics for observation) was associated with unfavorable outcomes in 3 studies when compared to alternative monitoring approaches, including medication self-administration [41], DOT by Anganwadi (community

health) workers [78], and 99DOTS (cellphone-based monitoring) [76]. Lack of community health worker support [73] or non-participation in TB support groups [126] was associated with unfavorable outcomes in 2 studies. Type of DOT or adherence monitoring approach had nonsignificant associations in 4 studies [75,84,105,109].

Suboptimal quality of care contributed to unfavorable outcomes. Inadequate counseling regarding treatment duration and the curability of TB was associated with unfavorable outcomes in 2 studies [41,105]. Incorrect or inadequate information from the provider had nonsignificant associations in 3 analyses [73,93,123]. Lack of TB drug availability [123], including during COVID-19-related lockdowns [93], was associated with unfavorable outcomes in 2 studies.

Across 15 studies surveying people who experienced loss to follow-up or medication non-adherence, health system-related reasons included lack of faith in treatment (reported by 5% [10/201] [114] to 25% [8/32] [92] in 2 studies), long distance to the clinic (reported by 1% [2/150] [117] to 21% [35/167] [87] across 6 studies [87,112,114,116,117,119,121]), and high treatment costs (reported by 17% [25/150] [117] to 30% [60/201] [114] across 3 studies [114,117,120]) (Fig 8). Healthcare provider barriers (reported by 1% [2/150] [117] to 21% [6/28] [119] across 3 studies [114,117,119]) included providers refusing treatment, advising people to stop treatment, or not cooperating with care. Non-availability of medications at the clinic, including during COVID-19-related lockdowns, was reported by 3% (6/201) [114] to 13% (7/55) [122] of people across 3 studies [114,116,122].

**Factors associated with unfavorable treatment outcomes among people with drug-resistant TB in regression analyses.** 14 studies reported adjusted analyses involving people with drug-resistant TB, of which 12 studies focused on RR or MDR TB and 2 studies on isoniazid mono-resistant TB. For RR or MDR TB, men had higher unfavorable outcomes in 4 studies [128–131] (Fig 9). Sex had nonsignificant associations in 4 studies [132–135]. Older age—more than 35 or 45 years or per each year increase—was associated with unfavorable outcomes in 4 studies [129,131,132,136]. Age had nonsignificant associations in 7 studies [128,130,133–135,137,138].

Regarding clinical factors, markers of prolonged or advanced disease—including prior TB treatment history [128,133,134], longer time to treatment [130], and cavitary lesions [128,137] —were associated with unfavorable outcomes in 5 studies. Prior TB treatment and cavitary disease had nonsignificant associations in 2 studies [135,137] and 1 study [135], respectively. In 2 studies, resistance to > = 5 drugs [132] or individual resistance to ofloxacin, streptomycin, or ethambutol [128] were associated with unfavorable outcomes.

Regarding comorbidities, alcohol use was associated with unfavorable outcomes in 2 studies [137,139]. Smoking was associated with unfavorable outcomes in 1 study [137] and had a nonsignificant association in 1 study [139]. Undernutrition—measured as low pre-treatment BMI or weight or non-improvement in weight or serum albumin with treatment—was associated with unfavorable outcomes in 5 studies [128,129,131,133,136]. Undernutrition had nonsignificant associations in 3 studies [134,135,140]. Other comorbidities associated with unfavorable outcomes included HIV [137] and anemia [140] in 1 study each; HIV had nonsignificant associations in 2 studies [128,134]. Medication nonadherence was associated with unfavorable outcomes in 2 studies [128,136]. In 1 study, longer exposure to a support package—involving counseling, nutritional supplements, and cash transfer—was associated with lower unfavorable outcomes [131].

In 1 study of people with isoniazid mono-resistant TB, men, people older than 40 years, people with HIV, or people who used alcohol or tobacco had higher unfavorable outcomes [141] (S4 Appendix, Table D).

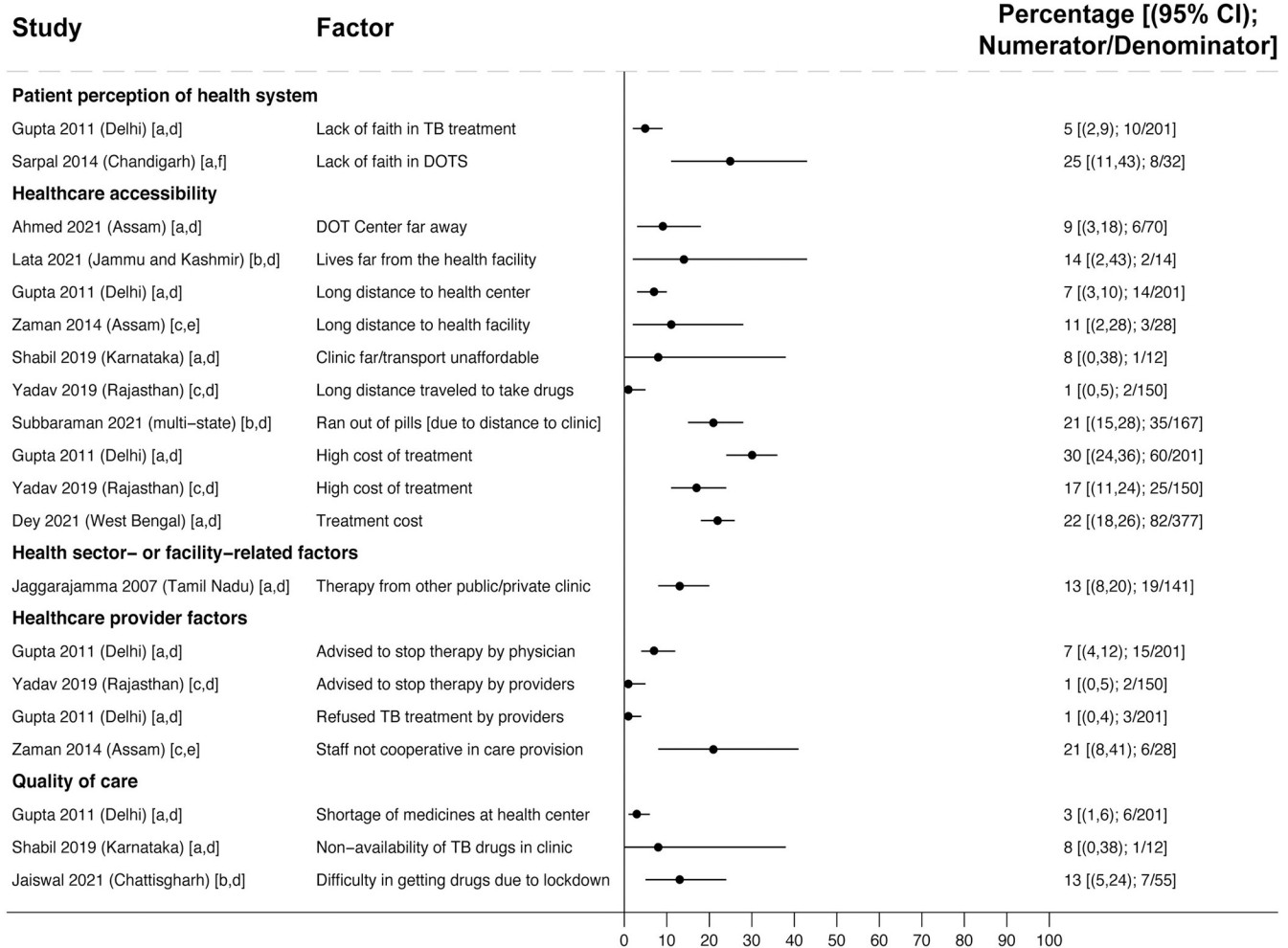

**Fig 8. Health system reasons for experiencing unfavorable treatment outcomes in people with drug-susceptible TB (Gap 4).** Studies report the estimated percentage of people interviewed who reported a given reason for experiencing unfavorable outcomes [87,92,112–114,116,117,119–122]. Study labels indicate outcomes are: [a] LTFU; [b] medication nonadherence; or [c] any interruption (defined as a combination of medication nonadherence and LTFU). Study labels indicate participants are: [d] from a combined population of people with new TB or a prior TB treatment history; [e] people with new TB only; [f] people with a prior TB treatment history only. CI, confidence interval; DOTS, directly observed therapy short course; LTFU, loss to follow-up; TB, tuberculosis.

**Reasons reported by people with drug-resistant TB for loss to follow-up or medication nonadherence during treatment.** Across 3 studies surveying people with RR or MDR TB who were lost to follow-up or experienced nonadherence, reasons included: migration out of the area (reported by 93% [26/28] of people in 1 study [142]), lack of family support (reported by 15% [18/122] of people in 1 study [143]), and medication adverse effects (reported by 7% [2/28] [142] to 75% [92/122] [143] of people across 3 studies [142–144]) (Fig M in the S6 Appendix).

**Factors associated with unfavorable TB treatment outcomes among people with HIV in regression analyses.** Across 5 studies reporting adjusted analyses among people with HIV, clinical factors associated with unfavorable treatment outcomes included pulmonary (versus extrapulmonary) TB in 2 studies [145,146], previous TB treatment history in 3 studies [145–147], and TB medication adverse effects in 1 study [148] (Fig N in the S6 Appendix). HIV-related factors associated with unfavorable treatment outcomes included low CD4 cell count in 2 studies [147,149], not being on HIV therapy in 1 study [145], and not taking

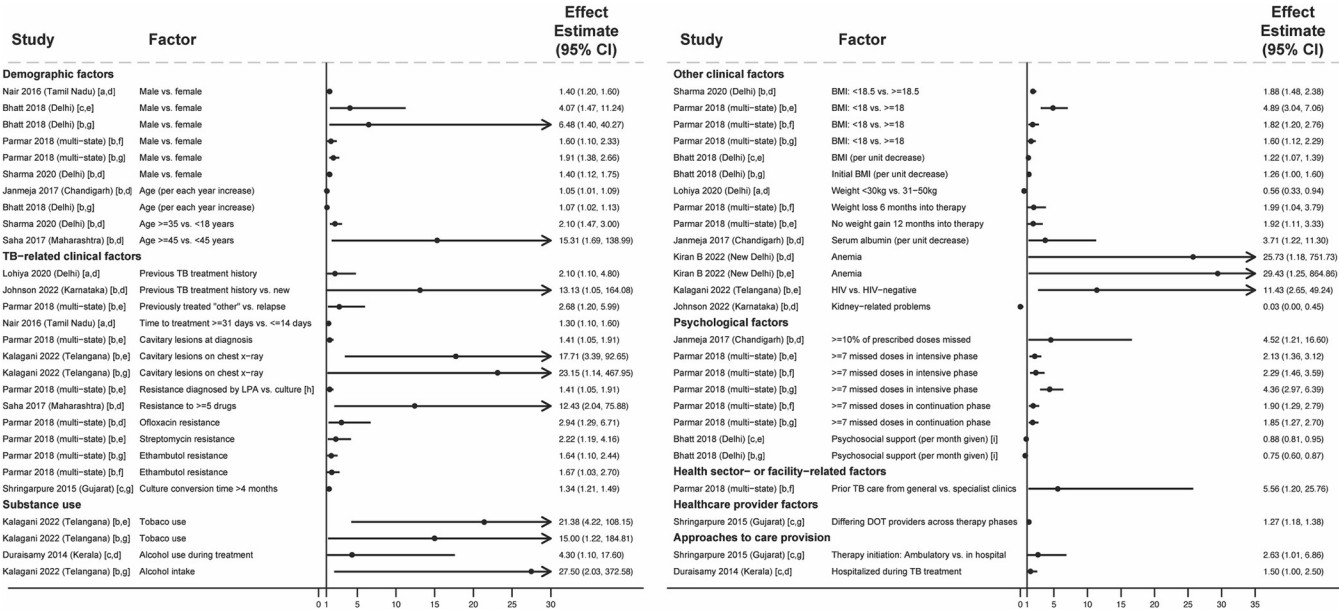

**Fig 9. Factors associated with unfavorable treatment outcomes in people with RR or MDR TB (Gap 4).** All studies used multivariable regression and reported adjusted effect estimates [128–137,139,140]. Estimates greater than 1 represent increased risk of unfavorable outcomes; estimates less than 1 represent decreased risk of unfavorable outcomes. Arrowheads means that the upper or lower limits of the CI extend beyond the end of the x-axis. Study labels indicate effect estimates are: [a] relative risk ratios; [b] odds ratios; or [c] hazard ratios. Study labels indicate outcomes are: [d] any unfavorable treatment outcome; [e] death; [f] treatment failure; or [g] loss to follow-up. Variable labels indicate: [h] higher mortality among LPA-diagnosed individuals may reflect survivor bias related to culture-diagnosed individuals dying before starting MDR TB treatment; and [i] psychosocial support package comprised nutritional supplementation, cash transfer, and counseling. Only statistically significant findings are presented. Some studies in the review with adjusted analyses reported nonsignificant findings for sex [132–135], age [128,130,133–135,137,138], previous TB history [135,137], cavitary disease [135], HIV status [128,134], BMI/weight [134,135,140], and tobacco use [139]. BMI, body mass index; CI, confidence interval; DOT, directly observed therapy; HIV, human immunodeficiency virus; kg, kilogram; LPA, line probe assay; MDR, multidrug-resistant; RR, rifampin-resistant; TB, tuberculosis.

cotrimoxazole prophylaxis in 2 studies [145,148]. Status of taking HIV antiretroviral therapy [147] and of taking cotrimoxazole therapy [146] had nonsignificant associations in 1 study each. In 1 study, nondisclosure of HIV status and lack of counseling before treatment were associated with unfavorable treatment outcomes [148].

**Factors associated with unfavorable treatment outcomes among children with TB.** In 1 study reporting an adjusted analyses including children with TB, extensively drug-resistant TB (versus MDR TB with less advanced resistance) and undernutrition (i.e., BMI-for-age less than 2 standard deviations below the average) were associated with unfavorable treatment outcomes [150] (Table D in the S4 Appendix).

## Gap 5—Barriers to achieving recurrence-free survival after TB treatment

**Factors associated with TB recurrence in regression analyses.** Across 9 studies reporting adjusted analyses involving TB recurrence, male sex was associated with TB recurrence in 2 studies [95,106] (Fig 10 and Table D in the S5 Appendix); sex had nonsignificant associations in 2 studies [81,111]. Medication nonadherence was associated with TB recurrence in 2 studies [95,151]; adherence had nonsignificant associations in 2 studies [81,106]. Posttreatment symptoms—measured by clinical evaluation or the Saint George's Respiratory Questionnaire (which assesses respiratory health in obstructive airways disease)—were associated with TB recurrence in 2 studies [152,153]. Low pretreatment BMI (alone [111] or with alcohol use [124]), and unimproved BMI after the intensive treatment phase [77], were associated with TB recurrence in 3 studies. Undernutrition had a nonsignificant association in 1 study [106].

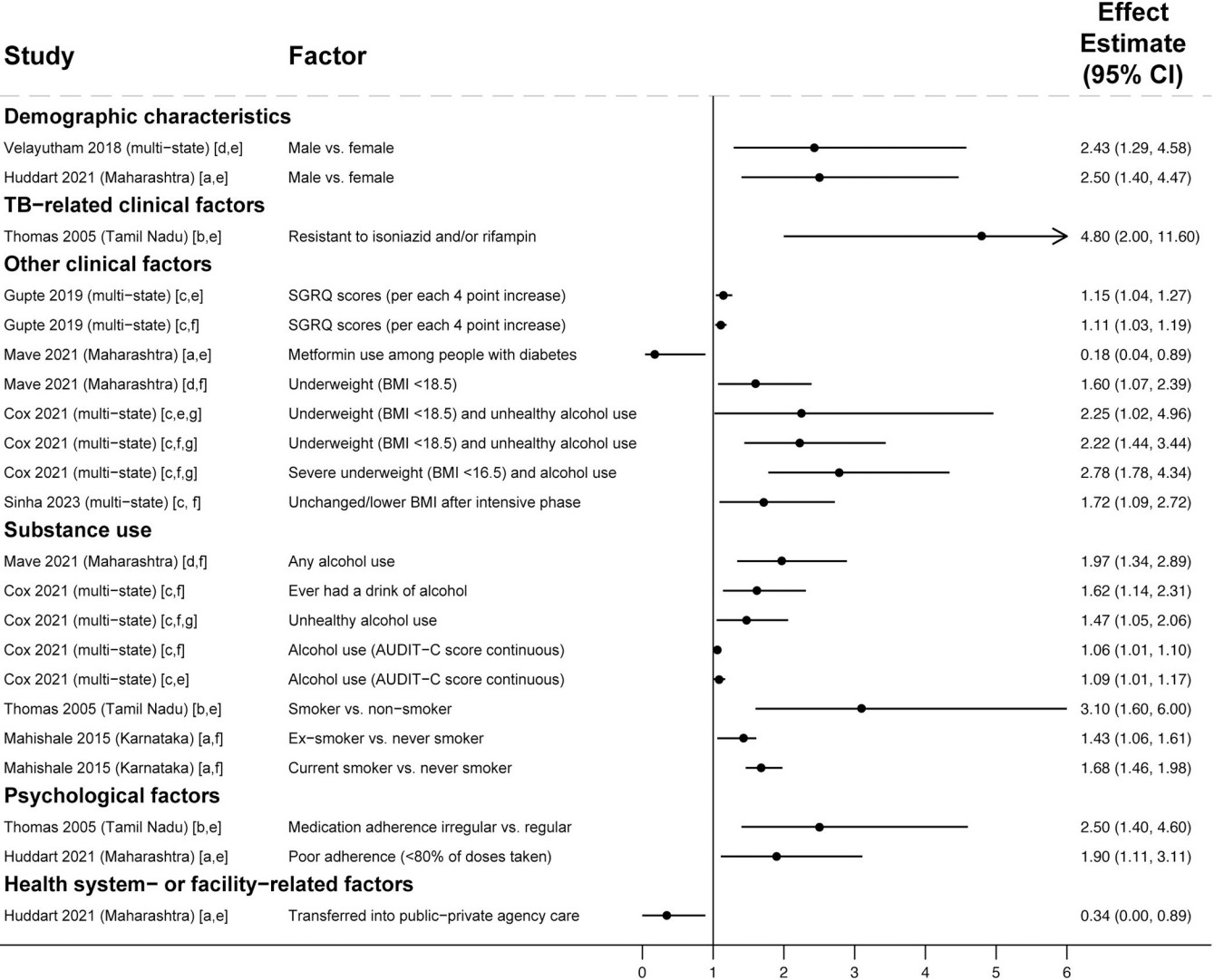

**Fig 10. Factors associated with TB recurrence after completing TB treatment as a single outcome or part of a composite outcome (Gap 5).** All studies used multivariable regression and report adjusted effect estimates [77,81,95,106,111,124,151,152,154]. Estimates greater than 1 represent increased risk of TB recurrence; estimates less than 1 represent decreased risk of recurrence. Arrowhead means that the upper limit of the CI extends beyond the end of the x-axis. Study labels indicate effect estimates are: [a] hazard ratios, [b] odds ratios, [c] incidence rate ratios, or [d] relative risk ratios. Other labels indicate: [e] study reported TB recurrence as a single outcome; [f] study reported TB recurrence as a composite outcome including on-treatment outcomes; and [g] unhealthy alcohol use was defined as AUDIT-C score > = 4. Only statistically significant findings are presented. Some studies in the review with adjusted analyses reported nonsignificant findings for sex [81,111], medication adherence [81,106], undernutrition [106], and smoking [106,111]. AUDIT, alcohol use disorder identification test; BMI, body mass index; CI, confidence interval; SGRQ, Saint George Respiratory Questionnaire; TB, tuberculosis.

Alcohol use was associated with TB recurrence in 2 studies [111,124]. Current or past smoking was associated with TB recurrence in 2 studies [151,154]; smoking had nonsignificant associations in 2 studies [106,111].

**Factors associated with posttreatment mortality in regression analyses.** Across 8 studies reporting adjusted analyses, male sex was associated with posttreatment mortality in 2 studies [155,156] (Fig 11 and Table D in the S5 Appendix). Sex had a nonsignificant association in 2 studies [81,95]. Older age—per year increase or greater than 25, 40, 44, or 60 years—was associated with posttreatment mortality in 5 studies [81,155–158]. Age had a nonsignificant association in 1 study [95]. Unemployment was associated with mortality in 2 studies [129,130].

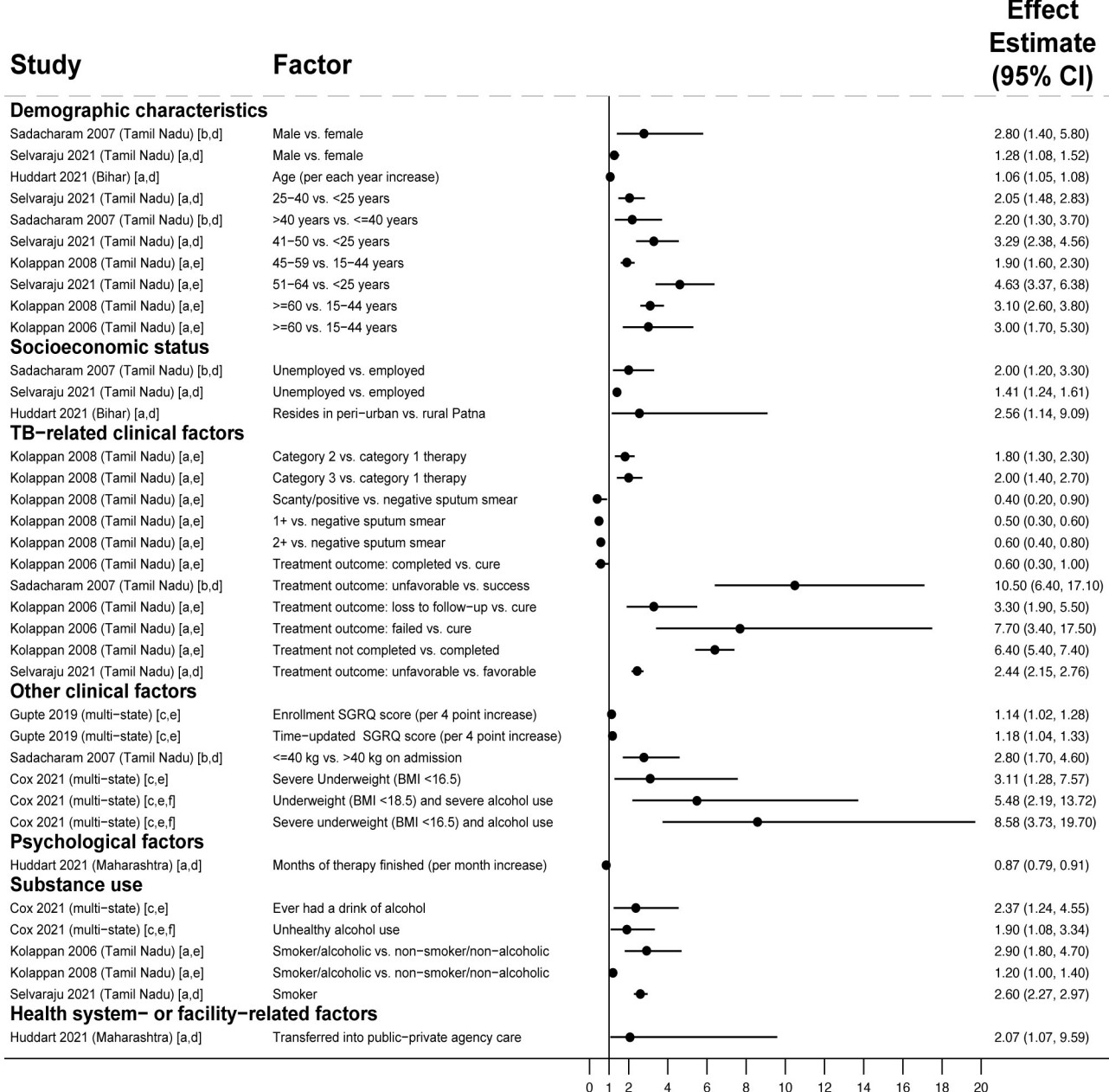

**Fig 11. Factors associated with mortality after TB treatment (without evaluation of TB recurrence) (Gap 5).** All studies used multivariable regression with findings reported as adjusted effect estimates [81,95,124,152,155–158]. Effect estimates greater than 1 represent increased mortality risk; estimates less than 1 represent decreased mortality risk. Study labels indicate: [a] effect estimates are hazard ratios; [b] effect estimates are odds ratios; [c] effect estimates are incidence rate ratios; [d] posttreatment mortality was reported as a single outcome; [e] posttreatment mortality was reported as part of a composite outcome including on-treatment mortality; and [f] unhealthy or severe alcohol use was defined as AUDIT-C score > = 4. Only statistically significant findings are presented. Some studies in the review with adjusted analyses reported nonsignificant findings for sex [81,95], age [95], and previous TB treatment history (i.e., treatment category) [81,95]. AUDIT, alcohol use disorder identification test; kg, kilogram; BMI, body mass index; SGRQ, Saint George Respiratory Questionnaire; TB, tuberculosis.

Previous TB treatment history (i.e., receiving category 2 therapy) was associated with post-treatment mortality in 1 study [158]; previous treatment history had nonsignificant associations in 2 studies [81,95]. Unfavorable on-treatment outcomes were associated with posttreatment mortality in 4 studies [155–158]. Similarly, increasing number of TB treatment

months was associated with lower posttreatment mortality in 1 study [95]; TB treatment months had a nonsignificant association in 1 study [81]. In 1 study, higher scores on the Saint George's Respiratory Questionnaire at TB diagnosis or during treatment were associated with posttreatment mortality [152]. Low pretreatment BMI was associated with posttreatment mortality in 1 study, when evaluated alone or with unhealthy alcohol use [124]. Low pretreatment weight ($<$ = 40 kilograms) was similarly associated with posttreatment mortality in 1 study [155]. Alcohol use was associated with posttreatment mortality in 1 study [124] and in combination with smoking in 2 studies [157,158]. Smoking was associated with posttreatment mortality in 1 study [153].

### Summary of common or important findings across care cascade gaps

We summarize factors that were statistically significantly associated with unfavorable outcomes for more than one care cascade gap, and the number of studies contributing findings, in Table B in the S6 Appendix. We summarize reasons for unfavorable outcomes reported across more than one gap, and the number of studies contributing findings, in Table C in the S6 Appendix. Fig 12 summarizes barriers that contributed to losses in each TB care cascade gap.

## Discussion

This systematic review, which synthesized more than 2 decades of studies, makes several important contributions to knowledge on care delivery across the TB care cascade in India [5]. First, our review highlights shortcomings of quantitative research conducted to date. Of concern across all care cascade gaps is the dearth of studies on children and the private sector, where about half of Indians with TB receive care [28,159]. In addition, few studies assessed health system factors, which are important to inform changes in care delivery.

Second, disaggregation of findings by care cascade gap and TB subpopulation provides granular information to program managers and researchers, who can use findings to inform intervention development for specific care gaps, subpopulations, or locations in India [12]. Third, when visualizing results, we clustered similar factors across studies, highlighting patterns of risk affecting TB outcomes. Programs might use these findings to prioritize high-risk subpopulations by providing greater attention and resources. For the remaining discussion, we consider common patterns of risk that emerged in this review.

### Continuities in risk across care cascade stages

Some TB subpopulations have higher risk of unfavorable outcomes across multiple care cascade stages. Men in the community were less likely to seek care for symptoms (Gap 1) [24,25] and to pursue TB evaluation after referral (Gap 2) [38]. These findings may explain the phenomenon of "missing men" in TB care [160], in which men are underrepresented in case notifications despite having higher TB population prevalence [161–163]. In several studies, men are also more likely to suffer unfavorable on-treatment and posttreatment outcomes (Gaps 4 and 5) [106,155]. TB services at every care stage should incorporate strategies to retain men. Older individuals—usually older than 40 years compared to younger age categories—were also more likely to suffer adverse outcomes across multiple gaps.

For people previously treated for TB, outcomes varied across gaps. On the one hand, they were more likely to pursue care for symptoms (Gap 1) [30] and TB evaluation when referred (Gap 2) [38], which may reflect better TB knowledge. On the other hand, they were more likely to experience PTLFU [40,57] and unfavorable treatment outcomes (Gaps 3 and 4), especially those who were lost to follow-up during their prior treatment [78,91,96,133,134]. These poor outcomes may reflect undiagnosed drug resistance, pulmonary disease from prior TB, or

| Gap 1<br>Not seeking care | Gap 2<br>Not completing the diagnostic workup | Gap 3<br>Pretreatment loss to follow-up | Gap 4<br>Not achieving treatment success | Gap 5<br>TB recurrence or post-treatment mortality |
|---|---|---|---|---|
| **Demographic factors**<br>• Male sex<br><br>**Socioeconomic barriers**<br>• Low socioeconomic status or income<br>• Financial constraints<br>• Work constraints<br>• Daily wage labor<br><br>**TB clinical barriers**<br>• Lower symptom severity or duration<br>• New TB (no prior TB)<br><br>**Substance use**<br>• Alcohol use<br>• Tobacco use<br><br>**Knowledge barriers**<br>• Lack of knowledge of symptoms, transmission, or curability of TB<br><br>**Health system barriers**<br>• Distrust of, or dissatisfaction with, local clinics / providers<br>• Healthcare provider indifference<br>• Long distance to clinics or lack of transport | **Demographic factors**<br>• Older age<br><br>**Socioeconomic barriers**<br>• Low socioeconomic status or income<br>• Financial constraints<br>• Work constraints<br><br>**TB clinical barriers**<br>• Lower symptom severity or duration or symptom improvement<br>• New TB (no prior TB)<br><br>**Substance use**<br>• Alcohol use<br><br>**Health system barriers**<br>• Not informed about concern for TB or need for further testing<br>• Negative healthcare provider interactions<br>• Long distance to diagnostic test (X-ray)<br>• Evaluated at tertiary or district vs. local facility<br>• Not identified as meeting NAAT / DST criteria (especially for smear-negative TB, extrapulmonary TB, TB recurrence, or prior loss to follow-up)<br>• Sputum sample lost during transfer to reference lab | **Demographic factors**<br>• Male sex<br>• Older age<br><br>**Socioeconomic barriers**<br>• Low socioeconomic status or education<br>• Work constraints<br>• Migration<br><br>**TB clinical barriers**<br>• Prior TB history<br>• Adverse effects during prior TB treatment<br>• Pre-treatment mortality<br><br>**Substance use**<br>• Alcohol use<br>• Tobacco use<br><br>**Psychological factors**<br>• Stigma<br>• Treatment refusal or disbelief in diagnosis<br><br>**Health system barriers**<br>• Distrust of, or dissatisfaction with, government clinics<br>• Diagnosis at private clinic/lab or high-volume government clinic<br>• Difficulty engaging with clinic-based DOT<br>• Long distance to clinics<br>• TB drug stockouts<br>• Person untraceable due to poorly recorded contact information | **Demographic factors**<br>• Male sex<br>• Older age<br><br>**Socioeconomic barriers**<br>• Lower education<br>• Work constraints<br>• Migration or travel<br><br>**TB clinical barriers**<br>• Longer illness duration<br>• Advanced disease<br>• Smear-positive disease<br>• Prior TB history (especially with prior loss to follow-up or treatment failure)<br>• Drug resistance<br>• TB drug adverse effects<br>• Early symptom improvement<br><br>**Other clinical barriers**<br>• HIV<br>• Undernutrition<br><br>**Substance use**<br>• Alcohol use<br>• Smoking<br><br>**Psychological barriers**<br>• Depression<br>• Medication nonadherence (e.g. forgetting to take drugs)<br><br>**Knowledge barriers**<br>• Lack of knowledge of TB therapy duration or curability<br><br>**Family-/society-related**<br>• Lack of family support<br>• Social stigma / discrimination<br><br>**Health system barriers**<br>• Lack of TB counseling<br>• Dissatisfied with TB services<br>• Difficulty engaging with clinic-based DOT<br>• Distance to clinics and lack of transportation<br>• Unavailability of TB drugs | **Demographic factors**<br>• Male sex<br>• Older age<br><br>**Socioeconomic barriers**<br>• Unemployment<br><br>**TB clinical barriers**<br>• Smear positive pulmonary disease<br>• Prior TB history<br>• Drug resistance<br>• Unfavorable on-treatment outcome<br><br>**Other clinical barriers**<br>• Undernutrition<br>• Post-treatment respiratory symptoms<br><br>**Substance use**<br>• Alcohol use<br>• Tobacco use<br><br>**Psychological factors**<br>• Medication nonadherence—both poor dosing implementation (day-to-day adherence) and non-persistence (inadequate treatment duration) |

**Fig 12. Important barriers to engagement in the care cascade for TB disease in India.** Barriers listed generally represent factors from regression analyses that were statistically significantly associated with unfavorable TB treatment outcomes in at least 2 studies for a given gap, or reasons that were reported by at least 15% of people with TB who experienced unfavorable outcomes in at least 1 study for a given gap. DOT, directly observed therapy; DST, drug susceptibility testing; NAAT, nucleic acid amplification testing; TB, tuberculosis.

continuation of behavioral risks that led to prior unfavorable outcomes. Previously treated people should be a focus of the NTEP's efforts to reduce PTLFU and unfavorable treatment outcomes.

Lower socioeconomic status was also associated with unfavorable outcomes across all care cascade gaps. Our findings also unpack how lower socioeconomic status shapes outcomes. Findings from multiple gaps suggest the association between lower education and unfavorable outcomes may relate to inadequate TB knowledge [24,26,58,109,114,115,117]. Work constraints were reported as a reason for poor outcomes in several studies, aligning with findings that daily wage laborers (who are not paid if they miss work) may be especially vulnerable [25,29,87]. Challenges related to patient mobility in several studies align with findings that migrant laborers [59,118] and people seeking care outside of their residential location [57,164] are more vulnerable to unfavorable outcomes. Structural barriers to reaching clinics—e.g., prohibitive distance, costs, or transportation—were another pathway by which socioeconomic

status contributed to unfavorable outcomes. While the NTEP provides direct benefits transfer (i.e., cash transfer) to people with TB [165], our findings suggest that a broader array of strategies is needed to support people experiencing poverty.

India's 2019 to 2021 TB prevalence survey found that half of people with TB in the population had no symptoms, and, among those with symptoms, two-thirds had not sought care, partly due to low symptom severity [163]. Gap 1 results similarly show that people with lower symptom duration or severity were less likely to have sought care [25,30], and people who had not sought care often reported low symptom severity as a reason [24,31–37]. In Gap 2, people with lower symptom severity or resolving symptoms were less likely to pursue or complete the diagnostic workup. In Gap 4, people reported mild symptoms or early symptom resolution as reasons for loss to follow-up in several studies. In addition, people diagnosed by active case finding—which identifies people at a less symptomatic stage—were more likely to experience pretreatment and on-treatment loss to follow-up (Gaps 3 and 4) [63,88]. Active case finding has individual and public health benefits; however, our review highlights challenges in retaining people diagnosed in this manner. Active case finding programs should consider counseling or incentives at each care stage to improve outcomes.

Alcohol use, smoking, and undernutrition are associated with higher TB prevalence [163] and were also associated with poor outcomes across multiple care cascade stages. Public health interventions—such as targeted nutritional support [166,167], higher alcohol and tobacco taxes, or effective implementation of the public smoking ban—may reduce TB incidence and improve care engagement. Nutritional support and counseling or medication-assisted therapy for alcohol use and smoking should be integrated into TB care to improve treatment outcomes and reduce TB recurrence.

Our findings highlight the need to improve TB care in India's public and private health sectors. People evaluated in the private sector were more likely to not pursue further workup (Gap 2) [40] or start TB treatment (Gap 3) [40,56]. This aligns with findings of a prior systematic review showing that initial contact with private sector providers was associated with greater delay in TB diagnosis in India [168]. At the same time, negative perceptions of local (often government) services were associated with poor outcomes or reported as barriers in several studies across Gaps 1 to 4, suggesting a need to improve the care experience in government services (e.g., polite provider behavior, shorter wait times [169]). Clinic-based DOT in the public sector, which requires that people go to clinics for observed dosing, contributed to PTLFU [41,62] and loss to follow-up from treatment [41,76,78,104]; however, use of clinic-based DOT has declined in India in recent years, partly due to a shift to daily dosing regimens (from thrice-weekly dosing) and more person-centered care models [75].

## Findings specific to each care cascade gap

Our review also highlights findings that are specific to each gap. Gap 1 findings indicate that TB knowledge motivates care-seeking [24–26], suggesting that mass communication regarding TB (e.g., on television, social media) may be an important intervention. For Gap 1, structural barriers to reaching clinics (e.g., work constraints, transportation barriers) may be addressed by bringing screening closer to people through active case finding. However, to be effective, active case finding initiatives must incentivize people who are hard-to-reach (e.g., daily wage laborers, people with low symptom severity) to engage in care.

For Gap 2, health system barriers had a major role in non-completion of the diagnostic workup. Lack of test accessibility—e.g., free chest X-rays or NAAT in local clinics—was a key obstacle [43,52]. Providers also missed identifying up to 54% of people who met criteria for NAAT [47], suggesting algorithms that target advanced diagnostic testing run the risk of

excluding eligible people [44]. Communication gaps were a barrier, because people were often not informed they were undergoing sputum testing or X-ray due to concern for TB [41,43,45].

For Gap 3, many people who experienced PTLFU (4% to 40%) died before starting treatment, likely from advanced TB [57,60–63]. By detecting TB early, active case finding may improve later care cascade outcomes. TB stigma was a barrier among 10% to 23% of people experiencing PTLFU [41,58,62], highlighting a need for robust counseling. Poor recording of contact information by providers increased PTLFU risk and led to difficulties tracing 10% to 52% of these patients [57,60,61,63]. Regular performance feedback on the readability and completeness of paper and electronic registers may reduce PTLFU [170].

For Gap 4, medication nonadherence was associated with poor treatment outcomes (Gap 4) and TB recurrence (Gap 5) in multiple studies [73,95,102,103,109,128,136], affirming that adherence is a crucial mediator of outcomes [15]. However, measuring adherence in routine care is difficult [171,172]. Use of novel and accurate approaches for detecting nonadherence, like urine drug metabolite testing [87,173], may facilitate early identification of people at risk for poor outcomes, so they can be given additional support. Medication adverse effects contributed to loss to follow-up across numerous studies for people with drug-susceptible TB (with up to 42% to 47% of people who stopped therapy doing so due to adverse effects [92,113,115]) and drug-resistant TB (with up to 75% of patients who stopped therapy doing so due to adverse effects [143]). Addressing adverse effects and other barriers, such as depression [93,94,101,110], TB stigma [66,89,92], and insufficient TB knowledge [109,114,115,117], will require integration of better counseling into routine care.

For Gap 5, posttreatment outcomes are shaped by the quality of care in earlier care cascade stages, and Gap 5 findings align with factors identified in earlier gaps, including male sex, older age, undernutrition, alcohol use, smoking, drug resistance, and medication nonadherence. Preventing posttreatment TB recurrence and death therefore depends on addressing risk factors upstream in the care cascade. At the same time, close posttreatment follow-up could facilitate early detection of new TB cases, which may be especially important given the poorer outcomes of people with previous TB.

## Strengths and limitations of the review

A strength of this review is our comprehensive approach to extracting and visualizing factors from regression analyses and reasons from surveys of people with TB, which allows triangulation of findings from both types of studies to identify patterns of risk, within and across care cascade gaps. Given that people with TB globally often come from socioeconomic disadvantage, our findings may have relevance for other high TB incidence countries. Our novel approach provides a roadmap for similar analyses to understand reasons for losses in the TB care cascade in other global settings.

A limitation is that we did not perform meta-analyses, because heterogeneity was high for most exposures and because of debates in the field of implementation science regarding whether findings from diverse real-world settings should be meta-analyzed [174]. We also did not assess publication bias, as meta-analyses were not performed in this review and publication bias is premised on the assumption that included studies were designed to assess the same prespecified hypotheses. Another limitation in visualization of findings is that we did not include statistically nonsignificant findings in our Forest plots due to space constraints; however, these nonsignificant findings are available for readers in Table D in each of the S1–S5 appendices. Any future meta-analyses conducted using our review findings should ensure inclusion of relevant nonsignificant findings that we reported.

While many findings have relevance at the national level, as India's NTEP oversees care for 2.1 million people notified with TB annually [2], our approach may raise concerns about

generalizability, given India's diverse population. Using causal transportability theory, program implementers can consider which risk factors identified in this review may be applicable to their setting to inform locally relevant interventions [175]. Our exclusion of qualitative findings from this review is also a limitation, as these studies provide insights that are often unobtainable from quantitative studies [169,176–178]. Finally, our approach to Gap 1 focused on care-seeking by people with TB symptoms; however, studies of TB diagnostic delays [168] and healthcare provider behavior using standardized patients [179–182] both suggest that health system barriers contribute substantially to Gap 1 losses. People developing interventions for Gap 1 should read these studies in parallel with our review findings.

This systematic review organized findings from 2 decades of studies on India's TB care cascade to illuminate patterns of risk shaping outcomes for people with TB disease. In addition to summarizing gap-specific findings, we identify findings contributing to unfavorable outcomes across multiple care cascade gaps. These factors included male sex, older age, poverty-related barriers, lower symptom duration and severity, undernutrition, alcohol use, smoking, and distrust of (or dissatisfaction with) health services. Closing gaps in the TB care cascade will reduce mortality, enhance well-being for people with TB, and curb TB transmission. Developing interventions to close these gaps, informed by this robust evidence regarding major risk factors, must therefore be central to India's ambitious plan to eliminate TB and should also inform investments by international funders to advance the global End TB agenda [1,10].

## Supporting information

**S1 Appendix. Methods and study characteristics for the systematic review of barriers to care seeking by individuals with TB symptoms in the community (Gap 1).**
(PDF)

**S2 Appendix. Methods and study characteristics for the systematic review of barriers to completion of the TB diagnostic workup (Gap 2).**
(PDF)

**S3 Appendix. Methods and study characteristics for the systematic review of barriers to treatment initiation for people diagnosed with tuberculosis disease (Gap 3).**
(PDF)

**S4 Appendix. Methods and study characteristics for the systematic review of barriers to achieving treatment success in people with TB disease (Gap 4).**
(PDF)

**S5 Appendix. Methods and study characteristics for the systematic review of barriers to achieving recurrence-free survival after completing tuberculosis treatment (Gap 5).**
(PDF)

**S6 Appendix. Extended results for all care cascade gaps.**
(PDF)

**S7 Appendix. PRISMA Checklist for all care cascade gaps.**
(PDF)

## Acknowledgments

We are grateful to Drs. Rajaram S. and Gururaj Patil for conducting secondary data analyses for Gap 1 from the Tuberculosis Health Action Learning Initiative project implemented by the Karnataka Health Promotion Trust, which was funded by the United States Agency for

International Development. We are grateful to Amy Lapidow and Paige Scudder, librarians at the Tufts Hirsh Health Sciences Library, for performing the refresher searches.

## Author Contributions

**Conceptualization:** Tulip A. Jhaveri, Pruthu Thekkur, Vineet Chadha, Srinath Satyanarayana, Ramnath Subbaraman.

**Data curation:** Tulip A. Jhaveri, Disha Jhaveri, Amith Galivanche, Maya Lubeck-Schricker, Dominic Voehler, Katherine Powers, Ramnath Subbaraman.

**Formal analysis:** Tulip A. Jhaveri, Disha Jhaveri, Maya Lubeck-Schricker, Mei Chung, Hemant Deepak Shewade, Ramnath Subbaraman.

**Funding acquisition:** Ramnath Subbaraman.

**Investigation:** Tulip A. Jhaveri, Disha Jhaveri, Amith Galivanche, Maya Lubeck-Schricker, Dominic Voehler, Mei Chung, Pruthu Thekkur, Vineet Chadha, Ruvandhi Nathavitharana, Ajay M. V. Kumar, Hemant Deepak Shewade, Katherine Powers, Kenneth H. Mayer, Jessica E. Haberer, Paul Bain, Madhukar Pai, Srinath Satyanarayana, Ramnath Subbaraman.

**Methodology:** Tulip A. Jhaveri, Mei Chung, Pruthu Thekkur, Vineet Chadha, Ruvandhi Nathavitharana, Ajay M. V. Kumar, Hemant Deepak Shewade, Katherine Powers, Kenneth H. Mayer, Jessica E. Haberer, Paul Bain, Madhukar Pai, Srinath Satyanarayana, Ramnath Subbaraman.

**Project administration:** Tulip A. Jhaveri, Ramnath Subbaraman.

**Resources:** Ramnath Subbaraman.

**Software:** Maya Lubeck-Schricker.

**Supervision:** Ramnath Subbaraman.

**Validation:** Tulip A. Jhaveri, Disha Jhaveri, Amith Galivanche, Dominic Voehler, Mei Chung, Katherine Powers, Ramnath Subbaraman.

**Visualization:** Tulip A. Jhaveri, Amith Galivanche, Maya Lubeck-Schricker, Mei Chung, Ramnath Subbaraman.

**Writing – original draft:** Tulip A. Jhaveri, Ramnath Subbaraman.

**Writing – review & editing:** Tulip A. Jhaveri, Disha Jhaveri, Amith Galivanche, Maya Lubeck-Schricker, Dominic Voehler, Mei Chung, Pruthu Thekkur, Vineet Chadha, Ruvandhi Nathavitharana, Ajay M. V. Kumar, Hemant Deepak Shewade, Katherine Powers, Kenneth H. Mayer, Jessica E. Haberer, Paul Bain, Madhukar Pai, Srinath Satyanarayana, Ramnath Subbaraman.

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
