## [Editor Report · Decision Letter 0]

17 May 2023

Dear Dr Subbaraman, 

Thank you for submitting your manuscript entitled "Barriers to engagement by people with tuberculosis disease in the care cascade in India: a systematic review of two decades of quantitative research" for consideration by PLOS Medicine.

Your manuscript has now been evaluated by the PLOS Medicine editorial staff and I am writing to let you know that we would like to send your submission out for external peer review.

Please re-submit your manuscript within two working days, i.e. by May 19 2023 11:59PM.

Kind regards,

Alexandra Schaefer, PhD

Associate Editor

PLOS Medicine

---

## [Decision Letter · Decision Letter 1]

28 Jul 2023

Dear Dr. Subbaraman,

Thank you very much for submitting your manuscript "Barriers to engagement by people with tuberculosis disease in the care cascade in India: a systematic review of two decades of quantitative research" (PMEDICINE-D-23-01329R1) for consideration at PLOS Medicine. 

Your paper was evaluated by an associate editor and discussed among all the editors here. It was also discussed with an academic editor with relevant expertise, and sent to independent reviewers, including a statistical reviewer. The reviews are appended at the bottom of this email and any accompanying reviewer attachments can be seen via the link below:

[LINK]

In light of these reviews, I am afraid that we will not be able to accept the manuscript for publication in the journal in its current form, but we would like to consider a revised version that addresses the reviewers' and editors' comments. Obviously we cannot make any decision about publication until we have seen the revised manuscript and your response, and we plan to seek re-review by one or more of the reviewers. 

We expect to receive your revised manuscript by Aug 16 2023 11:59PM. Please email us (plosmedicine@plos.org) if you have any questions or concerns.

We look forward to receiving your revised manuscript. 

Sincerely,

Alexandra Schaefer, PhD

PLOS Medicine

plosmedicine.org

PMEDICINE-D-23-01329R1

GENERAL

Please respond to all editor and reviewer comments detailed below in full.

Please cite the reference numbers in square brackets (e.g., “We used the techniques developed by our colleagues [19] to analyze the data”). Citations should be preceding punctuation.

Please cite your Supporting Information as outlined here: https://journals.plos.org/plosmedicine/s/supporting-information

Please ensure to be consistent in the use of numbers as words or numerals (e.g., ll.751-756).

Please remove the Financial disclosure statement and Competing interest statement following your ‘Acknowledgements’. This information should only be added in the according section in the online submission form.

ACADEMIC EDITOR COMMENTS

Clearly an ambitious article with some work remaining but has lots of potential.

EDITORIAL COMMENTS

We appreciate your efforts to streamline the manuscript. However, we still feel that the manuscript is quite long and that the manuscripts could benefit from an even greater focus on reader accessibility by possibly placing some data and information in the appendix.

We noticed that the Methods and Results sections contain quite a lot of discussion of methodological approaches - we appreciate that the study is very complex, but wonder how much of this could be included/reported as supporting information as part of an analysis plan? This could potentially provide more focus and further streamline your manuscript.

Finally, we noticed that the figures look very 'busy', especially Figure 6. Could the data presented in Figure 6 be presented in a table instead? What information value do the forest plots add, given that the confidence intervals are shown next to them? Please consider these points while we leave the modification of your figures to your discretion. 

COMPETING INTEREST STATEMENT

All authors must declare their relevant competing interests per the PLOS policy, which can be seen here:

https://journals.plos.org/plosmedicine/s/competing-interests

For authors with ties to industry, please indicate whether any of the interests has a financial stake in the results of the current study."

Please add this statement to the manuscript's Competing Interests: "MP is an Academic Editor on PLOS Medicine's editorial board."

ABSTRACT

Please report your abstract according to PRISMA for abstracts, following the PLOS Medicine abstract structure (Background, Methods and Findings, Conclusions) http://www.plosmedicine.org/article/info:doi/10.1371/journal.pmed.1001419 .

Please provide the dates of search, data sources, types of study designs included, eligibility criteria, and synthesis/appraisal methods. 

Abstract Methods and Findings:

* Please ensure that all numbers presented in the abstract are present and identical to numbers presented in the main manuscript text.

PLOS Medicine requests that main results are quantified with 95% CIs as well as p values. Please include. When reporting p values please report as p<0.001 and where higher as the exact p value p=0.002, for example. For the purposes of transparent data reporting, if not including the aforementioned please clearly state the reasons why not.

Please include any important dependent variables that are adjusted for in the analyses.

Throughout, suggest reporting statistical information as follows to improve clarity for the reader “22% (95% CI [13%,28%]; p</=)”. Please amend throughout the abstract and main manuscript.

Please note the use of commas to separate upper and lower bounds, as opposed to hyphens as these can be confused with reporting of negative values.

AUTHOR SUMMARY

Thank you for providing an Author Summary. Please shorten the individual bullet points. The summary should include 2-3 single sentence, individual bullet points under each of the questions. This text is subject to editorial change and should be distinct from the scientific abstract. In the final bullet point of ‘What Do These Findings Mean?’, please describe the main limitations of the study in non-technical language.

Please see our author guidelines for more information: https://journals.plos.org/plosmedicine/s/revising-your-manuscript#loc-author-summary.

It may be helpful to review currently published articles for examples which can be found on our website here https://journals.plos.org/plosmedicine

INTRODUCTION

Please address past research and explain the need for and potential importance of your study. Indicate whether your study is novel and how you determined that. If there has been a systematic review of the evidence related to your study (or you have conducted one), please refer to and reference that review and indicate whether it supports the need for your study. 

Please conclude the Introduction with a clear description of the study question or hypothesis.

METHODS AND RESULTS

Please report your SR/MA according to the PRISMA guidelines provided at the EQUATOR site.

http://www.equator-network.org/reporting-guidelines/prisma/

Please provide the completed PRISMA checklist and upload it as Supporting Information.

Please add the following statement, or similar, to the Methods: "This study is reported as per the Preferred Reporting Items for Systematic Reviews and Meta-Analyses (PRISMA) guideline (S1 Checklist)."

Please update your search to the present time.

Please evaluate evidence of publication bias.

Please mention the language of studies included in your systematic review and if not done so, please include non-English language sources of studies.

PLOS Medicine requests that main results are quantified with 95% CIs as well as p values. Please include. When reporting p values please report as p<0.001 and where higher as the exact p value p=0.002, for example. For the purposes of transparent data reporting, if not including the aforementioned please clearly state the reasons why not.

Please include any important dependent variables that are adjusted for in the analyses.

Suggest reporting statistical information as detailed above – see under ABSTRACT

ll.169-171: For the current manuscript, and since the focus is on the quantitative data reported in this study, should mention of the qualitative data be reserved for the future paper? We suggest changing these statements to "In the current study, we only report quantitative findings. Qualitative findings will be reported elsewhere. We discuss methodological considerations common to both the quantitative and qualitative elements of our study" or similar.

l.305: Please change to “This allows readers to identify trends (or discordant findings) across studies.”.

l.308: Please define ‘HIV’ at first use.

ll.308-311: Here, you describe that Table 2 summarizes the framework organizing findings into “people-, family, or society-related factors” and “health system-related factors,” with relevant subcategories. In the column header of Table 2, the specific factors listed are described as “Examples of factors that might be included in a subcategory”. Please provide a full overview of the specific factors included in the subcategories or adjust the column header if Table 2 provides the full set of factors.

l.453: Please define ‘DR’.

l.471: Please define ‘PTLFU’ at first use.

ll.474-475: Please add the according citation for the different age definitions.

l.511: Please define ‘DOT’.

ll.554-556: Please add the according citation for the different age definitions.

ll.682-684: Please change “People with HIV, or with unknown HIV status, had higher unfavorable outcomes in three studies [93,96,98], as was unknown diabetes status (versus not having diabetes) [96].” to “People with HIV or unknown HIV status had higher unfavorable outcomes in three studies [93,96,98], as had people with unknown diabetes status (versus not having diabetes) [96].”.

l.754: Please ensure to consistently use ‘kilograms’ or ‘kg’ throughout your manuscript. 

l.810: Some readers may not be familiar with the Saint George's Respiratory Questionnaire, please add some details to make it accessible to a wide audience.

DISCUSSION

Please present and organize the Discussion as follows: a short, clear summary of the article's findings; what the study adds to existing research and where and why the results may differ from previous research; strengths and limitations of the study; implications and next steps for research, clinical practice, and/or public policy; one-paragraph conclusion. Please remove any subheadings.

ll.870-872 suggest: “Of concern across all gaps in the care cascade is the dearth of studies on children and the private sector, where about half of Indians with TB receive care.”

ll.887-888 suggest: “Some TB subpopulations are at higher risk of unfavorable outcomes across multiple stages of the care cascade.” 

ll.887-888: At the end of the statement, it seems you mistakenly cited Table 3 and Table 4 instead of Table 4 and Table 5. Please revise.

l.1074: change ‘heath system’ to ‘health system’.

Table 4/5: Please remove Tables 4 and 5 from the Discussion, as new results should not be discussed first in the Discussion. We suggest moving these tables to the Appendix, given the length of your manuscript and the fact that the data presented appear to have been "summarized" earlier in your manuscript. In addition, we have found that the colors make the tables difficult to read. We suggest using asterisks instead of colors to indicate the study numbers included.

FIGURES

Please provide titles and legends for all figures (including those in Supporting Information files).

For all Figures, please ensure that you have complied with our figures requirements http://journals.plos.org/plosmedicine/s/figures.

Please consider avoiding the use of red and green in order to make your figure more accessible to those with colour blindness.

Please ensure to define abbreviations used in your figures.

Please indicate in the figure caption the meaning of the whiskers and the arrowhead.

For some figures, we noted that there is an overlap of text and whiskers (e.g., Figure 9). Please revise and ensure to have enough space for each column and/or add a line break when necessary (including the Figures in the Supporting Information files).

Figure 3: Please define ‘PTLFU’.

Figure 3/5/6/8/9/10: Please mark the vertical line as ‘1’ in your Figure or add ‘(vertical line)’ behind ‘1’ in your Figure description.

Figure 3/4/5/6/7/8/9/10: Please indicate the meaning of the information in brackets following the study (Author, year).

Figure 4: Please add a header for the third column as “%” is not sufficient.

Figure 5: For Factor ‘Age >=45 vs. <45’, please add the unit ‘years’.

Figure 

---

## [Decision Letter · Decision Letter 2]

27 Mar 2024

Dear Dr. Subbaraman,

Thank you very much for re-submitting your manuscript "Barriers to engagement in the care cascade for tuberculosis disease in India: a systematic review of quantitative studies" (PMEDICINE-D-23-01329R2) for review by PLOS Medicine.

Thank you for your detailed response to the editors' and reviewers' comments. I have discussed the paper with my colleagues and the academic editor, and it has also been seen again by three of the original reviewers. The changes made to the paper were mostly satisfactory to the reviewers. As such, we intend to accept the paper for publication, pending your attention to the editorial comments below in a further revision. When submitting your revised paper, please once again include a detailed point-by-point response to the editorial comments.

[LINK]

In revising the manuscript for further consideration here, please ensure you address the specific points made by each reviewer and the editors. In your rebuttal letter you should indicate your response to the reviewers' and editors' comments and the changes you have made in the manuscript. Please submit a clean version of the paper as the main article file. A version with changes marked must also be uploaded as a marked up manuscript file. Please also check the guidelines for revised papers at http://journals.plos.org/plosmedicine/s/revising-your-manuscript for any that apply to your paper.

We ask that you submit your revision within 1 week (Apr 03 2024). However, if this deadline is not feasible, please contact me by email, and we can discuss a suitable alternative.

Please do not hesitate to contact me directly with any questions (aschaefer@plos.org). If you reply directly to this message, please be sure to 'Reply All' so your message comes directly to my inbox.

We look forward to receiving the revised manuscript.

Sincerely,

Alexandra Schaefer, PhD

Associate Editor 

PLOS Medicine

plosmedicine.org

Requests from Editors:

GENERAL COMMENTS

1) Please include page numbers and line numbers in the manuscript file. Use continuous line numbers (do not restart the numbering on each page). For review purposes, we have added line numbers to the file “Main manuscript - Barriers to TB care cascade - Clean.docx” (attached).

2) We note that you have switched between the terms gender and sex throughout the manuscript. Please note that the terms gender and sex are not interchangeable (as discussed in https://www.who.int/health-topics/gender); please use the appropriate term (we suggest "sex").

ABSTRACT

l.2: Please define “TB” at first use.

AUTHOR SUMMARY

l.60: Please exchange “barriers” with “factors”.

METHODS AND RESULTS

1) In line with Reviewer #2's comments and after discussion with the Academic Editor, we ask you to include the nonsignificant results in the figures instead of showing only the significant results. You may add a footnote to the relevant results to indicate that these are non-significant results.

2) ll.147-162: Under "Search strategy and study selection", please include the initials of the two reviewers who assessed the articles for eligibility. Please also include the initials under "Data extraction".

3) We have noted that throughout the main text you use vague descriptions, such as “half of people”; “one-third of people” (l.287/l.289), which we feel will be difficult for the reader to imagine what these descriptions mean in terms of actual numbers. Could you provide actual event numbers in brackets? 

The same goes for Figures 2, 4, and 6. The percentages give little insight into the magnitude of the numbers presented (2% of 1000 people or 10000 makes a big difference). Therefore, we suggest adding the actual numbers (nominator/denominator). Please revise throughout the main manuscript.

4) l.318 ff: "(reported by 15% in 1 study [42])" (example) - Similar to the comment above, we suggest reporting the nominator and denominator. Please revise once again throughout the main manuscript.

5) Figure 6: Please define “DOT/DOTS”.

DISCUSSION

1) l.870: Please temper claims of primacy of results by stating, "to our knowledge" or something similar.

2) Please remove the "Conclusion" subheading. The Conclusion paragraph should be a continuous part of the Discussion section.

REFERENCES

1) Where website addresses are cited, please specify the date of access and use the word “accessed” instead of “cited” (e.g. [accessed: 12/06/2023]).

SOCIAL MEDIA

To help us extend the reach of your research, please provide any X (formerly known as Twitter) handle(s) that would be appropriate to tag, including your own, your co-authors’, your institution, funder, or lab. Please enter in the submission form any handles you wish to be included when we post about this paper.

Comments from Reviewers:

Reviewer #1: The authors have addressed all my points including the important one about non-statistically-significant effects.

Michael Dewey

Reviewer #2: It was a pleasure to read the updated version of this article. It has been greatly improved by the edits made, and strikes a balance between comprehensively and consistently reporting the entirety of the work done, while also providing a clear synthesis of the findings. 

My only remaining comment: while I appreciate the inclusion of the non-significant findings (also requested by Reviewer 1), I am not sure why it's not possible to also include these on the forest plots themselves (as opposed to just in the captions), to more accurately display the entirety of the data. I can see that this might be crowded but I worry that the current presentation may be misleading. To pick one example, I find it interesting that in most studies for Gap 4, educational attainment was not significantly associated with worse outcomes - not the impression given by Figure 5. Is this due to wide confidence intervals (suggesting that people with less education are indeed at consistently higher risk of LTFU and may need more tailored support) or because there was truly no effect (suggesting that in most settings education does not act as an obstacle to completing TB treatment)? This seems like exactly the kind of heterogeneity a forest plot is designed to show. This is a suggestion, and I defer to the statistical reviewer (Reviewer 1), editors and authors on this point.

Otherwise many thanks for the opportunity to review an interesting piece of work. 

Reviewer #4: The authors have addressed all my comments/concerns adequately

[LINK]

General Editorial Requests

---

## [Editor Report · Decision Letter 3]

29 Apr 2024

Dear Dr Subbaraman, 

On behalf of my colleagues and the Academic Editor, Amitabh Bipin Suthar, I am pleased to inform you that we have agreed to publish your manuscript "Barriers to engagement in the care cascade for tuberculosis disease in India: a systematic review of quantitative studies" (PMEDICINE-D-23-01329R3) in PLOS Medicine.

I appreciate your thorough responses to the reviewers' and editors' comments throughout the editorial process. We look forward to publishing your manuscript, and editorially there are only a few remaining minor stylistic/presentation points that should be addressed prior to publication. We will carefully check whether the changes have been made. If you have any questions or concerns regarding these final requests, please feel free to contact me at aschaefer@plos.org.

Please see below the minor points that we request you respond to:

1) l.189: Please change to: “2 reviewers (from TJ, DJ, AG, or DV) independently extracted..”

2) ll.862-864: Please provide a reference.

3) Figure 2/4/6/8: Thank you for including the numerator/denominator. We suggest changing the order in which the values are presented, i.e. Percentage ([95% CI]; Numerator/ Denominator). This will make it easier for readers to compare the percentages, while still having the actual numbers for full disclosure. For example: Figure 8, first row: 5 ([2,9]; 10/201). 

PRESS

Sincerely, 

Alexandra Schaefer, PhD 

Associate Editor 

PLOS Medicine